



# (GO)²-SIM: A GCM-Oriented Ground-Observation Forward-Simulator Framework for Objective Evaluation of Cloud and Precipitation Phase

Katia Lamer[1], Ann M. Fridlind.[2], Andrew S. Ackerman.[2], Pavlos Kollias[3,4,5],
Eugene E. Clothiaux[1] and Maxwell Kelley[2]

[1] Department of Meteorology and Atmospheric Science, Pennsylvania State University, University Park, 16802, U.S.A.
[2] NASA Goddard Institute for Space Studies, New York, 10025, U.S.A.
[3] Environmental & Climate Sciences Department, Brookhaven National Laboratory, Upton, 11973, U.S.A.
[4] School of Marine and Atmospheric Sciences, Stony Brook University, Stony Brook, 11794, U.S.A.
[5] University of Cologne, Cologne, 50937, Germany

*Correspondence to*: Katia Lamer (kxl5431@psu.edu)

## Abstract

General circulation model (GCM) evaluation using ground-based observations is complicated by inconsistencies in hydrometeor and phase definitions. Here we describe (GO)²-SIM, a forward-simulator designed for objective hydrometeor phase evaluation, and assess its performance over the North Slope of Alaska using a one-year GCM simulation. For uncertainty quantification, 18 empirical relationships are used to convert model grid-average hydrometeor (liquid and ice, cloud and precipitation) water contents to zenith polarimetric micropulse lidar and Ka-band Doppler radar measurements producing an ensemble of 576 forward-simulation realizations. Sensor limitations are represented in forward space to objectively remove from consideration model grid cells with undetectable hydrometeor mixing ratios, some of which may correspond to numerical noise.

Phase classification in forward space is complicated by the inability of sensors to measure ice and liquid signals distinctly. However, signatures exist in lidar-radar space such that thresholds on observables can be objectively estimated and related to hydrometeor phase. The proposed phase classification technique leads to misclassification in fewer than 8% of hydrometeor-containing grid cells. Such misclassifications arise because, while the radar is capable of detecting mixed-phase conditions, it can mistake water- for ice-dominated layers. However, applying the same classification algorithm to forward-simulated and observed fields should generate hydrometeor phase statistics with similar uncertainty. Alternatively, choosing to disregard how sensors define hydrometeor phase leads to frequency of occurrence discrepancies of up to 40%. So, while hydrometeor phase maps determined in forward space are very different from model "reality" they capture the information sensors can provide and thereby enable objective model evaluation.



## 1 Introduction

The effect of supercooled water on the Earth's top-of-atmosphere energy budget is a subject of increasing interest owing to its wide variability across climate models and its potential impact on predicted equilibrium climate sensitivity (Tan et al., 2016; McCoy et al., 2016; Frey et al., 2017). Some general circulation models (GCMs) now prognose number concentrations and mass mixing ratios for both cloud and precipitation hydrometeors of both liquid and ice phase, which enables them to shift towards more realistic microphysical process-based phase prediction (e.g., Gettelman and Morrison, 2015; Gettelman et al., 2015). While more complete and physically sound, these models still contain multiple scheme choices and tuning parameters, creating a need for increasingly thorough evaluation and adjustment (e.g., Tan and Storelvmo, 2016; English et al., 2014).

Active remote sensing observations remain an indirect approach to evaluate models because they measure hydrometeor properties different from those produced by microphysical schemes. For each hydrometeor species within a grid cell models prognose geophysical quantities such as mass and number concentration, whereas active remote sensors measure power backscattered from all hydrometeors species present within their observation volumes. Defining which hydrometeors have an impact is a fundamental question that needs to be addressed by the modeling, as well as observational, communities. In numerical models it is not uncommon to find very small hydrometeor mixing ratio amounts as demonstrated below. They may possibly be unphysical, effectively numerical noise, and the decision of which hydrometeor amounts are physically meaningful is somewhat arbitrary. Considering sensor capabilities is one path to objectively assessing hydrometeor populations within models. On such a path it is possible to evaluate those simulated hydrometeor populations that lead to signals detectable by sensors, leaving unassessed those not detected. Sensor detection capabilities are both platform- and sensor-specific. Space-borne lidars can adequately detect liquid clouds globally but their signals cannot penetrate thick liquid layers, limiting their use to a subset of single-layer systems or upper-level cloud decks (Hogan et al., 2004). Space-borne radar observations, while able to penetrate multi-layer cloud systems, are of coarser vertical resolution and of limited value near the surface owing to ground interference and low sensitivity (e.g., Huang et al., 2012b; Battaglia and Delanoë, 2013; Huang et al., 2012a). A perspective from the surface can therefore be more appropriate for the study of low-level cloud systems (e.g., de Boer et al., 2009; Dong and Mace, 2003; Klein et al., 2009; Intrieri et al., 2002).

Fortunately, both sensor sampling and hydrometeor scattering properties can be emulated through the use of forward-simulators. Forward-simulators convert model output to quantities observed by sensors and enable a fairer comparison between model output and observations; discrepancies can then be more readily attributed to dynamical and microphysical differences rather than methodological bias. For example, the CFMIP (Cloud Feedback Model Intercomparison Project) Observation Simulator Package (COSP) is composed of a number of satellite-oriented forward-simulators (Bodas-Salcedo et al., 2011), including a lidar backscattering forward-simulator that has been used to evaluate the representation of upper-level supercooled water layers in GCMs (e.g., Chepfer et al., 2008; Kay et al., 2016). Also, Zhang et al. (2017) present a first attempt at a ground-based radar reflectivity simulator tailored for GCM evaluation.

Here we propose to exploit the complementarity of ground-based vertically pointing polarimetric lidar and Doppler radar measurements, which have been shown uniquely capable of documenting water phase in shallow and multi-layered cloud conditions near the surface where supercooled water layers frequently form. More specifically, we present a GCM-oriented ground-based observation forward-simulator [$(GO)^2$-SIM] framework designed for objective hydrometeor phase evaluation (Fig. 1). GCM output variables (Sec. 2) are converted to observables in three steps: 1) hydrometeor backscattered power estimation (Sec. 3), 2) consideration for sensor capabilities (Sec. 4) and, 3) estimation of specialized observables (Sec. 5). These



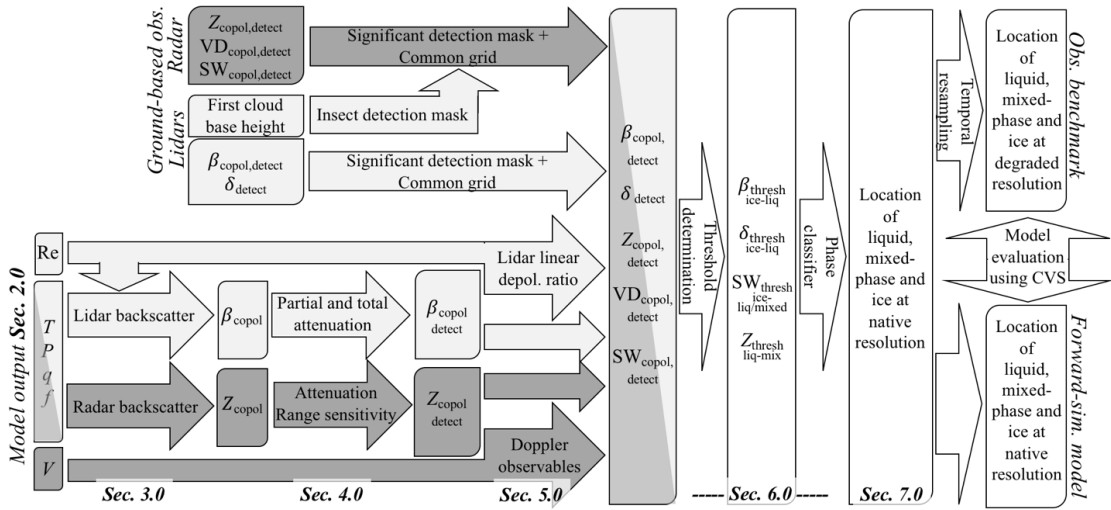

**Figure 1.** (GO)$^2$-SIM framework. (GO)$^2$-SIM treats/emulates two types of remote sensors: Ka-band Doppler radars (dark gray shading) and 532 nm polarimetric lidars (light gray shading). It then tunes and applies a common phase-classification algorithm (white boxes) to both observed (upper section) and forward-simulated (bottom section) fields. Follow-on work will show how an approach based on cloud vertical structure (CVS) can be used as a final step before model evaluation.

forward-simulated fields, similar to observed fields, are used as inputs to a multi-sensor water phase classifier (Sec. 6). The performance of (GO)$^2$-SIM is evaluated over the North Slope of Alaska using output from a one-year simulation of the current development version of the NASA Goddard Institute for Space Studies GCM, hereafter referred to by its generic name, ModelE. Limitations are discussed in Sec. 6.3 and uncertainty quantified in Sec. 7.

## 2   GCM Outputs Required as Inputs to the Forward-Simulator

      To demonstrate how atmospheric model variables are converted to observables we performed a one-year global simulation using the current development version of the ModelE GCM. Outputs from a column over the North Slope of Alaska (column centered at latitude 71.00° and longitude -156.25°) are input to (GO)$^2$-SIM. The most relevant changes from a recent version of ModelE (Schmidt et al. 2014) are implementation of the Bretherton and Park (2009) moist turbulence scheme and the Gettelman and Morrison (2015) microphysics scheme for stratiform cloud. The implementation of a two-moment microphysics scheme with prognostic precipitation species makes this ModelE version more suitable for the forward simulations presented here than previous versions. Here ModelE is configured with a 2.0° by 2.5° latitude-longitude grid with 62 vertical layers. The vertical grid varies with height from 10 hPa layer thickness over the bottom 100 hPa of the atmosphere, coarsening to about 50 hPa thickness in the mid-troposphere, and refining again to about 10 hPa thickness near the tropopause. For the current study, model top is at 0.1 hPa, though we limit our analysis to pressures greater than 150 hPa. Dynamics (large scale advection) is computed on a 225-s time step and column physics on a 30-min time step. High time-resolution outputs (every column physics time step) are used as input to (GO)$^2$-SIM. ModelE relies on two separate schemes to prognose the occurrence of stratiform and convective clouds. The current study focuses on stratiform clouds because their properties are more thoroughly diagnosed in this model version; when performing future model evaluation, the contribution from convective clouds will also be considered.





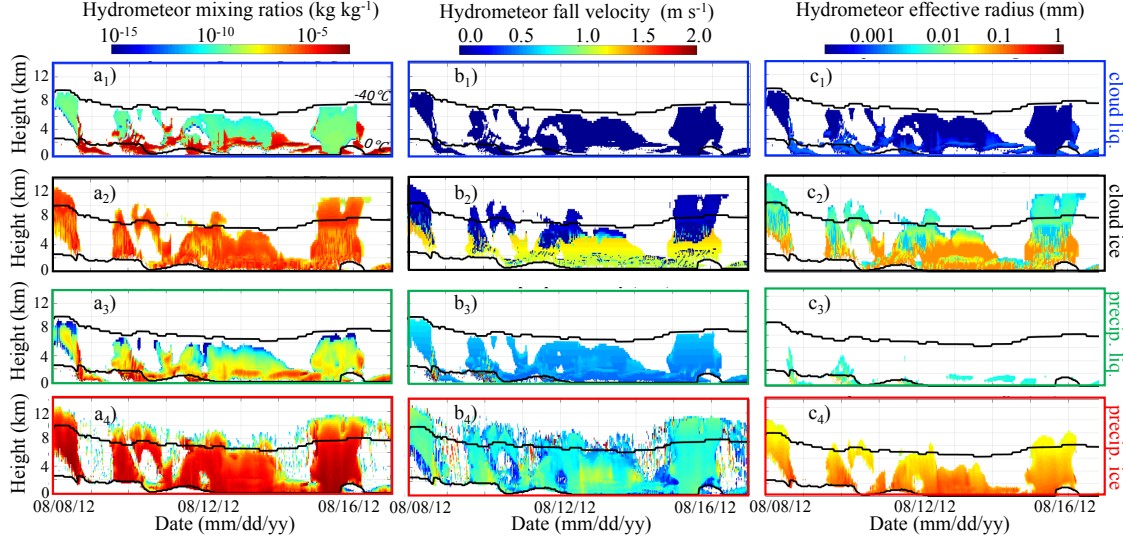

**Figure 2.** Sample time series of ModelE outputs: $a_{1-4}$) mixing ratios, $b_{2-4}$) mass weighted fall speed
(positive values indicate downward motion) and $c_{1-4}$) effective radii for cloud droplets (1; blue boxes),
cloud ice particles (2; black boxes), precipitating liquid drops (3; green boxes) and precipitating ice
particles (4; red boxes). Also indicated are the locations of the 0 °C and -40 °C isotherms (horizontal
black lines).

An example of eight days of this simulation is displayed in Fig. 2. From a purely numerical modelling
standpoint, the simplest approach to defining hydrometeors is to consider any nonzero hydrometeor mixing
ratio as physically meaningful. Using this approach, we find that 43.5 % of the 981,120 grid cells simulated
in the one-year ModelE run contain hydrometeors, with 2.4 % of them being pure liquid, 37.8 % pure ice
and 59.8 % mixed in phase (Table 1a). However, these statistics are impacted by a number of simulated
small hydrometeor mixing ratio amounts that may or may not result from numerical noise (e.g., Fig. 2a;
blue-green colors). The forward-simulator framework will be used to create phase statistics of only those
hydrometeors present in amounts that can create signal detectable by sensors hence removing the need for
arbitrary filtering.

(GO)²-SIM forward-simulator inputs are, at model native resolution, mean grid box temperature and
pressure as well as hydrometeor mixing ratios, area fractions (used to estimate in-cloud values), mass
weighted fall speeds and effective radii for four hydrometeor species: cloud liquid water, cloud ice,
precipitating liquid water and precipitating ice. In its current setup, (GO)²-SIM can accommodate any
model that produces these output variables
## 3  Hydrometeor Backscattered Power Simulator


Reaching a common objective hydrometeor definition between numerical model output and active
sensors starts by addressing the fact that they are based on different hydrometeor properties (i.e., moments).
Backscattering amounts, observed by sensors, depend on both sensor frequency and on hydrometeors
properties and amounts. Hydrometeor properties that impact backscattering include hydrometeor size,
phase, geometrical shape, orientation and bulk density. In most GCMs, however, such detailed information
is highly simplified (e.g., spherical ice particles) or not explicitly represented (e.g., orientation), which
introduces uncertainty in the process of transforming model output to observables.



**Table 1.** a) Hydrometeor phase frequency of occurrence obtained a) from ModelE mixing ratios outside of the forward-simulator framework, b) and c) from the forward simulation ensemble created using different backscattered power assumptions. The median and interquartile range (IQR) capture the statistical behavior of the ensemble. Results using thresholds b) objectively determined for each forward ensemble member, c) modified from those in Shupe (2007). Percentage values are relative either to the total number of simulated hydrometeor-containing grid cells (426,603) or those grid cells with detectable hydrometeor amounts (333,927). Note that the total number of simulated grid cells analyzed is 981,120.

| a) Determined using ModelE Output Hydrometeor Mixing Ratios | | | |
|---|---|---|---|
| | Grid cells containing only liquid phase | Grid cells containing mixed phase | Grid cells containing only ice phase | Simulated hydrometer-containing grid cells |
| Frequency of Occurrence (%) | 2.4 | 59.8 | 37.8 | 43.5 |

| b) Determined Using Flexible Objective Thresholds from Model Output Mixing-Ratios | | | | | | | | |
|---|---|---|---|---|---|---|---|---|
| | Grid cells classified as liquid phase | | Grid cells classified as mixed phase | | Grid cells classified as ice phase | | Grid cells containing detectable hydrometeors | |
| | Median | ½ IQR | Median | ½ IQR | Median | ½ IQR | Median | ½ IQR |
| Frequency of Occurrence (%) | 11.3 ± | 0.6 | 19.2 ± | 1.8 | 68.8 ± | 3.1 | 78.3 ± | 1.8 |
| False Positive (%) | 0.5 ± | 0.0 | 1.1 ± | 0.3 | 0.0 ± | 0.0 | 1.7 ± | 0.3 |
| False Negative (%) | 0.2 ± | 0.0 | See questionable row | | 1.5 ± | 0.2 | 1.7 ± | 0.3 |
| Questionable (%) | 1.4 ± | 0.0 | | | 3.8 ± | 0.9 | 5.2 ± | 0.9 |
| Total Error (%) | | | | | | | 6.9 ± | 1.1 |

| c) Determined Using Fixed Thresholds Modified from Shupe (2007) | | | | | | | | |
|---|---|---|---|---|---|---|---|---|
| | Grid cells classified as liquid phase | | Grid cells classified as mixed phase | | Grid cells classified as ice phase | | Grid cells containing detectable hydrometeors | |
| | Median | ½ IQR | Median | ½ IQR | Median | ½ IQR | Median | ½ IQR |
| Frequency of Occurrence (%) | 12.5 ± | 0.4 | 13.1 ± | 2.4 | 71.5 ± | 3.7 | 78.2 ± | 1.8 |
| False Positive (%) | 0.5 ± | 0.0 | 0.3 ± | 0.0 | 0.1 ± | 0.0 | 0.9 ± | 0.0 |
| False Negative (%) | 0.1 ± | 0.0 | See questionable row | | 0.7 ± | 0.0 | 0.9 ± | 0.0 |
| Questionable (%) | 1.4 ± | 0.0 | | | 5.3 ± | 1.1 | 6.7 ± | 1.1 |
| Total Error (%) | | | | | | | 7.6 ± | 1.1 |

### 3.1 Lidar Backscattered Power Simulator

At a lidar wavelength of 532 nm, backscattered power is proportional to total particle cross section per unit volume. Owing to their high number concentrations, despite their small size, cloud particles backscatter this type of radiation the most.

In the Cesana and Chepfer (2013) backscattering-simulator, Mie theory is used to convert hydrometeor effective radius to backscattered power; in such an approach cloud particles (both liquid and ice) are assumed spherical. To avoid having to rely on this assumption for ice particles, (GO)²-SIM employs empirical relationships. We adopt the Hu et al. (2007b) representation of liquid cloud extinction derived from CALIPSO and CERES-MODIS observations and retrievals of liquid water content and effective





radius (Table 2, Eq. 1). For cloud ice water content, a number of empirical relationships with lidar extinction have been proposed for various geophysical locations and ice cloud types using a variety of assumptions. Four of these empirical relationships are implemented in $(GO)^2$-SIM (Table 2, Eqns. 2-5 and references therein). These relationships will be used to create an ensemble of forward simulations that will be used for uncertainty quantification (see Sec. 7). Using these empirical relationships, a given water content can be mapped to a range of lidar extinction values (Fig. 3a). This spread depends both on the choice of empirical relationships and on the variability of the atmospheric conditions that affect them (i.e., atmospheric temperature and hydrometeor effective radius variability). Fig. 3a also illustrates the fundamental idea that lidar extinction increases with increasing water content and that for a given water content cloud droplets generally lead to higher lidar extinction than cloud ice particles.

Lidar co-polar backscattered power ($\beta_{copol,species}$ [$m^{-1}sr^{-1}$]) generated by each hydrometeor species is related to lidar extinction ($\sigma_{copol,species}$ [$m^{-1}$]) through the lidar ratio:

$$\beta_{copol,cl} = (1/18.6 \text{ sr}) \, \sigma_{copol,cl}. \qquad \text{(O'Connor et al., 2004) (6)}$$
$$\beta_{copol,ci} = (1/25.7 \text{ sr}) \, \sigma_{copol,ci}. \qquad \text{(Kuehn et al., 2016) (7)}$$

Of course, lidars do not measure cloud droplet backscattering independently of cloud ice particle backscattering. Rather they measure total co-polar backscattered power ($\beta_{copol,total}$) which the sum of the contribution from both cloud phases.

**3.2 Radar Backscattered Power Simulator**

At the cloud-radar wavelength of 8.56 mm (Ka-band), backscattered power is approximately related to the sixth power of the particle diameter, and inversely proportional to the forth power of the wavelength. Hereafter radar backscattered power will be referred to as "radar reflectivity" as commonly done in literature.

For reference, the COSP (Bodas-Salcedo et al., 2011) and ARM Cloud Radar Simulator for GCM (Zhang et al., 2017) packages both use QuickBeam for the estimation of radar reflectivity (Haynes et al., 2007). QuickBeam computes radar reflectivity using Mie theory and assuming all hydrometeor species to be spherical and of a specified density.

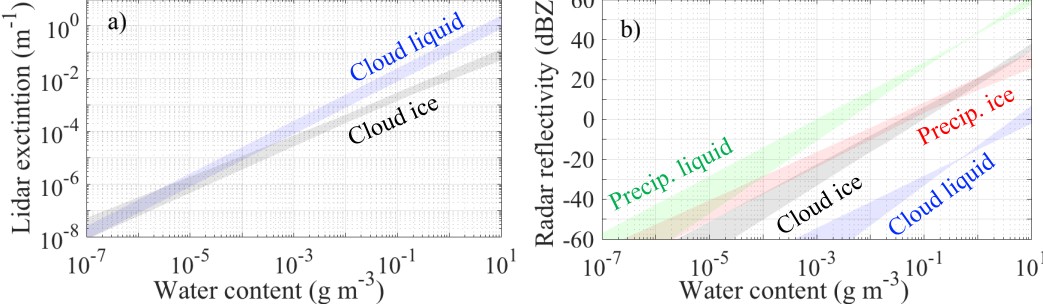

**Figure 3.** a) Lidar extinction as a function of water content in the form of water cloud (blue) and ice cloud (black). b) Radar co-polar reflectivity as a function of water content in the form of water cloud (blue) and precipitation (green) and ice cloud (black) and precipitation (red). Spread emerges from using multiple differing empirical relationships (listed in Table 2) and from variability in the one-year ModelE output (including the effects of varying temperature and effective radii).



**Table 2.** Empirical relationships used to convert hydrometeor water content (WC [g m$^{-2}$]) to lidar
extinction ($\sigma$ [m$^{-1}$]) and radar reflectivity ($Z$ [mm$^6$ m$^{-3}$]).

| Type | Eq. # | Relationships for lidar extinction | References |
|---|---|---|---|
| Cloud liq. (cl) | 1 | $\sigma_{\text{copol,cl}} = \dfrac{\text{WC}_{\text{cl}}(3/2)}{\text{Re } \rho_{\text{liq}}}$ with $\rho_{\text{liq}} = 1$ | Hu et al. (2007b) |
| Cloud ice (ci) | 2 | $\sigma_{\text{copol,ci}} = \left(\dfrac{\text{WC}_{\text{ci}}}{119}\right)^{1/1.22}$ | Heymsfield et al. (2005) |
| | 3 | $\sigma_{\text{copol,ci}} = \left(\dfrac{\text{WC}_{\text{ci}}}{a_3}\right)^{1/b_3}$ with $a_3 = 89 + 0.6204T$ and $b_3 = 1.02 - 0.0281T$ | Heymsfield et al. (2005) |
| | 4 | $\sigma_{\text{copol,ci}} = \left(\dfrac{\text{WC}_{\text{ci}}}{527}\right)^{1/1.32}$ | Heymsfield et al. (2014) |
| | 5 | $\sigma_{\text{copol,ci}} = \left(\dfrac{\text{WC}_{\text{ci}}}{a_2}\right)^{1/b_2}$ with $a_2 = 0.00532 * (T + 90)^{2.55}$ and $b_2 = 1.31e^{(0.0047T)}$ | Heymsfield et al. (2014) |

| Type | Eq. # | Relationships for radar reflectivity | References |
|---|---|---|---|
| Cloud liq. (cl) | 8 | $Z_{\text{copol,cl}} = 0.048\,\text{WC}_{\text{cl}}^{2.00}$ | Atlas (1954) |
| | 9 | $Z_{\text{copol,cl}} = 0.03\,\text{WC}_{\text{cl}}^{1.31}$ | Sauvageot and Omar (1987) |
| | 10 | $Z_{\text{copol,cl}} = 0.031\,\text{WC}_{\text{cl}}^{1.56}$ | Fox and Illingworth (1997) |
| Cloud ice (ci) | 11a | $Z_{\text{copol,ci}} = 10^{\left(\frac{\log_{10}(\text{WC}_{\text{ci}})+1.70+0.0233\,T}{0.072}/10\right)}$ | R. J. Hogan et al. (2006) |
| | 12 | $Z_{\text{copol,ci}} = \left(\dfrac{\text{WC}_{\text{ci}}}{0.064}\right)^{\frac{1}{0.58}}$ | Atlas et al. (1995) |
| | 13 | $Z_{\text{copol,ci}} = \left(\dfrac{WC_{\text{ci}}}{0.097}\right)^{\frac{1}{0.59}}$ | Liu and Illingworth (2000) |
| | 14 | $Z_{\text{copol,ci}} = \left(\dfrac{\text{WC}_{\text{ci}}}{0.037}\right)^{\frac{1}{0.696}}$ | Sassen (1987) |
| Precip. liq (pl) | 15 | $Z_{\text{copol,pl}}[\text{mm}^6\,\text{m}^{-3}] = \left(\dfrac{\text{WC}_{\text{pl}}}{0.0034}\right)^{\frac{7}{4}}$ | Hagen and Yuter (2003) |
| | 16 | $Z_{\text{copol,pl}}[\text{mm}^6\,\text{m}^{-3}] = \left(\dfrac{\text{WC}_{\text{pl}}}{0.0039}\right)^{\frac{1}{0.55}}$ | Battan (1973) |
| | 17 | $Z_{\text{copol,pl}} = \left(\dfrac{\text{WC}_{\text{pl}}}{0.00098}\right)^{\frac{1}{0.7}}$ | Sekhon and Srivastava (1971) |
| Precip. ice (pi) | 11b | $Z_{\text{copol,pi}} = 10^{\left(\frac{\log_{10}(\text{WC}_{\text{pi}})+1.70+0.0233\,T}{0.072}/10\right)}$ | R. J. Hogan et al. (2006) |
| | 18 | $Z_{\text{copol,pi}} = \left(\dfrac{\text{WC}_{\text{pi}}}{0.0218}\right)^{\frac{1}{0.79}}$ | Liao and Sassen (1994) |
| | 19 | $Z_{\text{copol,pi}} = \left(\dfrac{\text{WC}_{\text{pi}}}{0.04915}\right)^{\frac{1}{0.90}}$ | Sato et al. (1981) |
| | 20 | $Z_{\text{copol,pi}} = \left(\dfrac{\text{WC}_{\text{pi}}}{0.05751}\right)^{\frac{1}{0.736}}$ | Kikuchi et al. (1982) |



In contrast, (GO)²-SIM relies on water content-based empirical relationships to estimate cloud liquid water
(cl), cloud ice (ci), precipitating liquid water (pl) and precipitating ice (pi) radar reflectivity. It is expected
that these empirical relationships capture at least part of the impacts of hydrometeor non-sphericity and
inhomogeneity. A number of empirical relationships link hydrometeor water content to co-polar radar
reflectivity. Thirteen of these empirical relationships are implemented in (GO)²-SIM (Table 2, Eqns. 8-20
and references therein). These relationships are used to create an ensemble of forward simulations for
uncertainty quantification (see Sec. 7). Figure 3b illustrates the fact that for all these empirical relationships
increasing water content leads to increasing radar reflectivity. As already mentioned, radar reflectivity is
approximately related to the sixth power of the particle diameter, which explains why, for the same water
content, precipitating hydrometeors are associated with greater reflectivity than cloud hydrometeors.
In reality, radars cannot isolate energy backscattered by individual hydrometeor species. Rather they
measure total co-polar reflectivity ($Z_{\text{copol,total}}$ [mm$^6$ m$^{-3}$]) which is the sum of the contributions from all
of the hydrometeor species.
## 244  4  Sensor Capability Simulator
In the previous section, total backscattered power resulting from all modeled hydrometeor species
(without any filtering) is estimated. In order to objectively assess model hydrometeor properties, they must
be converted to quantities that are comparable to observations, necessitating incorporation of sensor
detection limitations, including attenuation and finite sensitivity. Fortunately, lidar and radar sensors are
often relatively well-characterized so that sensor detection capabilities can be quantified and replicated in
forward-simulators for an objective model-to-observation comparison.
### 253  4.1 Lidar Detection Capability
Following the work of Cesana and Chepfer (2013), the (GO)²-SIM lidar forward-simulator takes into
consideration that lidar power is attenuated by clouds. Attenuation is related to cloud optical depth ($\tau$),
which is a function of total cloud extinction ($\sigma_{\text{copol,total}}$ [m$^{-1}$]) that includes the effect of cloud liquid
water and cloud ice via:
$\tau = \int_{z0}^{z} \sigma_{\text{copol,total}} \text{dh}$ ,                                                                                                          (21)
Lidar attenuation is exponential and two-way as it affects the lidar power on its way out and back:
$\beta_{\text{copol,total,att}} = \beta_{\text{copol,total}} \, e^{-2\tau}$.                                                                                          (22)
In the current simulator we assume that only cloud segments with optical depth smaller than three can be
penetrated, other clouds being opaque (Cesana and Chepfer, 2013) such that total co-polar backscattered
power detected ($\beta_{\text{copol,total,detect}}$) is:
$\beta_{\text{copol,total,detect}} = \beta_{\text{copol,total,att}}$     where $\tau \leq 3$;
$\beta_{\text{copol,total,detect}} = \text{undetected}$     where $\tau > 3$.                                                                 (23)
For the sample ModelE output shown in Fig. 2, Fig. 4a illustrates results from the lidar forward-simulator
for one forward-ensemble member (i.e., using a single set of lidar backscattered power empirical
relationships specifically eqns. (1) and (4)). Figure 4a₁ shows lidar total co-polar backscattered power
without consideration of sensor limitations, such as attenuation, which are included in Fig. 4a₂. Lidar
attenuation prevents the tops of deep systems containing supercooled water layers from being observed



(e.g., magenta boxes on 08/10 and 08/13). For the one-year sample the forward-simulated lidar system
detects only 35.5% of simulated hydrometeor-containing grid cells. In Sec. 6 we will determine which
hydrometeors (liquid water or ice) are responsible for the detected signals.

**4.2 Radar Detection Capability**

284       Millimeter-wavelength radars are also affected by signal attenuation. Radar signal attenuation depends
both on the transmitted wavelength and on the mass and phase of the hydrometeors. Liquid phase
hydrometeors attenuate radar signals at all millimeter radar wavelengths, even leading to total signal loss in
heavy rain conditions. In contrast, water vapor attenuation is less important at relatively longer wavelengths
(e.g., 8.56 mm; the wavelength simulated here) but can be important near wavelengths of 3.19 mm (the
CloudSat operating wavelength; (Bodas-Salcedo et al., 2011)).

At 8.56 mm (Ka-band) total co-polar attenuated reflectivity ($Z_{copol,total,att}$ [dBZ]) is given by

$$Z_{copol,total,att} = Z_{copol,total} - a_{cl+pl}, \tag{24a}$$

where two-way liquid attenuation ($a_{cl+pl}$ [dB]) is estimated using cloud and precipitating liquid water
contents ($WC_{cl}$ and $WC_{pl}$ [g m$^{-3}$]) and the thickness of the liquid layer (dh [m]):

$$a_{cl+pl} = 2 \int_{z=0}^{z} \left[0.6 \left(WC_{pl} + WC_{cl}\right)\right] dh. \tag{24b}$$

In addition to attenuation, radars suffer from having a finite sensitivity that decreases with distance. Given
this, the total co-polar reflectivity detectable ($Z_{copol,total,detect}$ [dBZ]) is

$$Z_{copol,total,detect} = Z_{copol,total,att} \text{ where } Z_{copol,total,att} \geq Z_{min},$$
$$Z_{copol,total,detect} = \text{Undetected} \text{ where } Z_{copol,total,att} < Z_{min}, \tag{25a}$$

where the radar minimum detectable signal ($Z_{min}$ [dBZ]) is a function of height ($h$ [km]) and can be
expressed as

$$Z_{min} = Z_{sensitivity\ at\ 1\ km} + 20 \log_{10} h. \tag{25b}$$

A value of $Z_{sensitivity\ at\ 1\ km}$ = -41 dBZ is selected to reflect the sensitivity of the Ka-band ARM Zenith
Radar (KAZR) currently installed at the Atmospheric Radiation Measurement (ARM) North Slope of
Alaska observatory. This value has been determined by monitoring two years of observations and it reflects
the minimum signal observed at a height of 1 km. The minimum detectable signal used in the simulator
should reflect the sensitivity of the sensor used to produce the observational benchmark to be compared to
the forward-simulator output.

For the sample ModelE output shown in Fig. 2, Figure 4b illustrates results from the radar forward-
simulator for one forward-ensemble member (i.e., using a single set of radar reflectivity empirical
relationships specifically eqns. (9), (11a), (15) and (11b)). Figure 4b$_1$ shows radar total co-polar reflectivity
without consideration of sensor limitations, while Fig. 4b$_2$ includes the effects of attenuation and the range-
dependent minimum detectable signal. Sensor limitations make it such that heavy rain producing systems
cannot be penetrated (e.g., magenta box on 08/08 and 08/10) and the tops of deep systems cannot be
observed (e.g., red box on 08/15). For the one-year sample the forward-simulated radar system could detect
only 69.9 % of the simulated hydrometeor-containing grid cells. In Sec. 6 we will determine the phase of
the hydrometeors responsible for the detected signals.





**Figure 4.** Example outputs from the $(GO)^2$-SIM backscattered power modules (1), sensor capability modules (2) and specialized-observables modules (3-4) for a) lidars and b) radars obtained using one set of empirical backscattered power relationships. This figure highlights sensor limitations ranging from attenuation (magenta boxes) to sensitivity loss with range (red boxes). Also indicated are the locations of the 0 °C and -40 °C isotherms (black lines). Note that positive velocities indicate falling motion.





### 4.3 Lidar-Radar Complementarity

Figures 4a$_2$ and 4b$_2$ highlight the complementarity of lidar and radar sensors. Despite sensor limitations, 532 nm lidar measurements can be used to characterize hydrometeors near the surface and infer the location of a lowermost liquid layer if one exists. In contrast, 8.56 mm radars have the ability to penetrate cloud layers and light precipitation, allowing them to determine cloud boundary locations (e.g, Kollias et al., 2016). For the one-year sample ModelE output the combination of both sensors enables detection of 73.0 % of the hydrometeor-containing grid cells. Real observations can be used to objectively evaluate these detectable hydrometeor populations while nothing can be said about those that are not detectable. Note that a number of undetectable grid cells only contain trace amounts of hydrometeors, which could be the result of numerical noise. As such the approach of considering sensor detection limitations helps objectively remove numerical noise from consideration and allows model and observations to converge towards a common hydrometeor definition for a fair comparison.

### 5 Forward Simulation of Specialized Observables

In the previous section total co-polar backscattered powers are used to determine which simulated hydrometeors are present in sufficient amounts to be detectable by sensors hence removing numerical noise from consideration. However, determining the phase of the detectable hydrometeor populations can be achieved with much greater accuracy by using additional observables.

Backscattered power alone provides a sense of hydrometeor number concentration (from lidar) and hydrometeor size (from radar), but it does not contain information about hydrometeor shape nor does it provide any hint on the number of coexisting hydrometeor species, both of which are relevant for phase determination. However, such information is available from lidar depolarization ratios and radar Doppler spectral widths.

### 5.1 Lidar Depolarization Ratio Simulator

So far we have described how hydrometeors of all types and phases affect co-polar radiation. It is important to note that radiation also has a cross-polar component which is only affected by nonspherical particles. Ice particles, which tend to be nonspherical, are expected to affect this component while we assume that cloud droplets, which tend to be spherical, do not. Taking the ratio of cross-polar to co-polar backscattering thus provides information about the dominance of ice particles in a hydrometeor population. This ratio is referred to as the linear depolarization ratio ($\delta_{\text{detect}}$) and it can be estimated where hydrometeors are detected by the lidar.

$$\delta_{\text{detect}} = \frac{\beta_{\text{crosspol,ci,detect}}}{\beta_{\text{copol,total,detect}}}. \tag{26a}$$

According to Cesana and Chepfer (2013) analysis of CALIPSO observations, cloud ice particle cross-polar backscattering ($\beta_{\text{crosspol,detect}}$ [$m^{-1}sr^{-1}$]) can be approximated using the following relationship:

$$\beta_{\text{crosspol,ci,detect}} = \frac{0.29}{1-0.29}\, \beta_{\text{copol,ci,detect}}. \tag{26b}$$

For the sample ModelE output shown in Fig. 2, Fig. 4a$_3$ shows an example of forward-simulated lidar linear depolarization ratios estimated using one set of backscattered power empirical relationships.





### 5.2 Radar Doppler Moment Simulator

Specialty Doppler radars have the capability to provide information about the movement of hydrometeors in the radar observation volume. This information comes in the form of the radar Doppler spectrum, which describes how backscattered power is distributed as a function of hydrometeor velocity (Kollias et al., 2011). The zeroth moment of the Doppler spectral distribution (the spectral integral) is radar reflectivity, the first moment (the spectral mean) is mean Doppler velocity (VD) and the second moment (the spectral spread) is Doppler spectral width (SW). Rich information is provided by the velocity spread (i.e., SW) of the hydrometeor population including information regarding the number of coexisting species, turbulence intensity and spread of the hydrometeor particle size distributions. Typically, the effects of turbulence and hydrometeor size variations on the velocity spread for a single species are much smaller than the effect of mixed-phase conditions. As such, Doppler spectral width is a useful parameter for hydrometeor phase identification.

Forward-simulations of Doppler quantities have been performed for cloud models using bin microphysics (e.g., Tatarevic and Kollias, 2015) but not, to our knowledge, for GCMs using 2-moment microphysics schemes. Co-polar mean Doppler velocity and co-polar Doppler spectral width are subject to the same detection limitations as radar reflectivity. In fact, just like radar reflectivity, these observables are strongly influenced by large hydrometeors; that is, they are reflectivity-weighted velocity averages.

Our approach begins by quantifying the contribution of each species present ($P_{\text{species}}$), which is determined by the species detected co-polar reflectivity ($Z_{\text{copol,species,detect}}$ [mm$^6$ m$^{-3}$]) relative to the total detected co-polar reflectivity ($Z_{\text{copol,total,detect}}$ [mm$^6$ m$^{-3}$]):

$$P_{\text{species}} = \frac{Z_{\text{copol,species,detect}}}{Z_{\text{copol,total,detect}}}, \tag{27a}$$

together with

$$Z_{\text{copol,species,detect}} = Z_{\text{copol,species}} - a_{\text{cl+pl}}, \text{ where } Z_{\text{copol,total,att}} \geq Z_{\min}. \tag{27b}$$

In Eqns. 27a-b the subscript "species" represents cl, ci, pl, or pi. Attenuation ($a_{\text{cl+pl}}$) and minimum detectable signal ($Z_{\min}$) are as in Eq. 24. The mass-weighted fall velocity of each species ($V_{\text{species}}$ [m s$^{-1}$]) is a GCM output and the total mean Doppler velocity detected (VD$_{\text{copol,detect}}$ [m s$^{-1}$]) is the sum of the reflectivity-weighted contribution of each species:

$$\text{VD}_{\text{copol,detect}} = P_{\text{cl}}V_{\text{cl}} + P_{\text{pl}}V_{\text{pl}} + P_{\text{ci}}V_{\text{ci}} + P_{\text{pi}}V_{\text{pi}}. \tag{28}$$

Total Doppler spectral width (SW$_{\text{copol,detect}}$ [m s$^{-1}$]) is more complex and combines hydrometeor species fall velocity ($V_{\text{species}}$ [m s$^{-1}$]) and spectral width (SW$_{\text{species}}$ [m s$^{-1}$]) information:

$$\text{SW}_{\text{copol,detect}} =$$
$$P_{\text{cl}}\left(\text{SW}_{\text{cl}}^2 + \left(V_{\text{cl}} - \text{VD}_{\text{copol,detect}}\right)^2\right) + P_{\text{pl}}\left(\text{SW}_{\text{pl}}^2 + \left(V_{\text{pl}} - \text{VD}_{\text{copol,detect}}\right)^2\right) +$$
$$P_{\text{ci}}\left(\text{SW}_{\text{ci}}^2 + \left(V_{\text{ci}} - \text{VD}_{\text{copol,detect}}\right)^2\right) + P_{\text{pi}}\left(\text{SW}_{\text{pi}}^2 + \left(V_{\text{pi}} - \text{VD}_{\text{copol,detect}}\right)^2\right), \tag{29}$$

where the spectral widths of individual species are assigned climatological values. These climatological values are SW$_{\text{cl}} = 0.10$ m s$^{-1}$, SW$_{\text{ci}} = 0.05$ m s$^{-1}$, SW$_{\text{pi}} = 0.15$ m s$^{-1}$ and SW$_{\text{pl}} = 2.00$ m s$^{-1}$ (Kalesse et al., 2016).



For the sample ModelE output shown in Fig. 2, Figs. 4b$_3$ and 4b$_4$ respectively show examples of forward
simulated mean Doppler velocity and Doppler spectral width estimate using one set of empirical radar
reflectivity relationship.
**6   Water Phase Classifier Algorithm**
433          From a purely numerical modeling perspective the simplest approach to defining the phase of a
hydrometeor population contained in grid cells is to consider that any nonzero hydrometeor mixing ratio
species contributes to the phase of the population. Using this approach, in the one-year sample, we find that
the detectable hydrometeor-containing grid cells are 2.4 % pure liquid, 19.4 % pure ice and 78.2 % mixed
phase (Note how these water phase statistics differ by up to 18.4 % from Sec. 2 where all grid cells,
potentially including numerical noise, were considered). But determining hydrometeor phase in
observational space is not as straightforward. It is complicated by the fact that sensors do not record ice-
and liquid-hydrometeor returns separately but rather record total backscattering from all hydrometeors.
Retrieval algorithms are typically applied to the observed total backscattering to determine the phase of
hydrometeor populations. However, phase classification algorithms have limitations that require each
hydrometeor species to be present not only in nonzero amounts but in amounts sufficient to produce a
phase signal. Thus, hydrometeor phase statistics obtained from a numerical model in the absence of a
forward simulator are not necessarily comparable with equivalent statistics retrieved from observables,
especially in instances where one hydrometeor species dominates the grid cell and other species are present
in trace amounts. A common hydrometeor phase definition must be established to objectively evaluate the
phase of simulated hydrometeor populations using observations, which requires the development of a phase
classification algorithm that can be applied to observables both forward-simulated and real.
The scientific literature contains a number of phase classification algorithms with different levels of
complexity. Hogan et al. (2003) used regions of high lidar backscattered power as an indicator for the
presence of liquid droplets. Lidar backscattered power combined with lidar linear depolarization ratio has
been used to avoid some of the misclassifications encountered when using backscattered power alone (e.g.,
Yoshida et al., 2010; Hu et al., 2007a; Hu et al., 2009; Hu et al., 2010; Sassen, 1991). Hogan and O'Connor
(2004) proposed using lidar backscattered power in combination with radar reflectivity. While the
combination of radar and lidar backscattered powers is useful for the identification of mixed-phase
conditions, their combined extent remains limited to single layer clouds or to lower cloud decks because of
lidar signal attenuation. Shupe (2007) proposed a technique in which radar Doppler velocity information is
used as an alternative to lidar backscattering information (for ranges beyond that of lidar total attenuation)
to infer the presence of supercooled water in multi-layer systems. Figure 5 displays cartoons of Doppler
spectra that have the same total co-polar radar reflectivity but different total mean Doppler velocities (VD)
and Doppler spectral widths (SW) resulting from different hydrometeor species and combinations, thus
highlighting the added value of Doppler information. The contribution of each species to the total co-polar
reflectivity is indicated as a percentage in the top right of each subpanel. These scenarios show that VD
tends to be relatively small for pure liquid cloud (Fig. 5a$_6$), pure ice cloud (Fig. 5a$_2$), and even mixed-phase
non-precipitating cloud (Fig. 5a$_3$,a$_5$,b$_3$) and only tends to increase when precipitation is present in cloud
(Fig. 5 a$_4$,b$_3$,b$_4$,b$_5$) or below cloud (Fig. 5a$_1$,b$_2$), making VD a seemingly robust indicator for precipitation
occurrence but not for phase identification. These scenarios also show that SW tends to be relatively small
in single-phase clouds without precipitation (Fig. 5a$_2$,a$_6$), pure precipitating ice (Fig. 5a$_1$) and multi-species
clouds with a dominant hydrometeor species (Fig. 5a$_3$,a$_5$). On the other hand, SW tends to be large when
liquid precipitation is present (Fig. 5b$_1$,b$_2$,b$_5$) and in mixed-phase clouds without a dominant species (Fig.
5b$_3$,b$_4$,b$_5$). These scenarios suggest that large spectral widths are useful indicators for the presence of
supercooled rain and mixed-phase conditions. Scenarios where this interpretation of spectrum width is
incorrect will be discussed in Sec. 6.3.



Regardless of which observation they are based-on, the aforementioned phase classification schemes all
rely on assumption that hydrometeor phases when projected on observational space (e.g., lidar
backscattered power against lidar depolarization ratio) create well-defined patterns that can be separated
using thresholds.

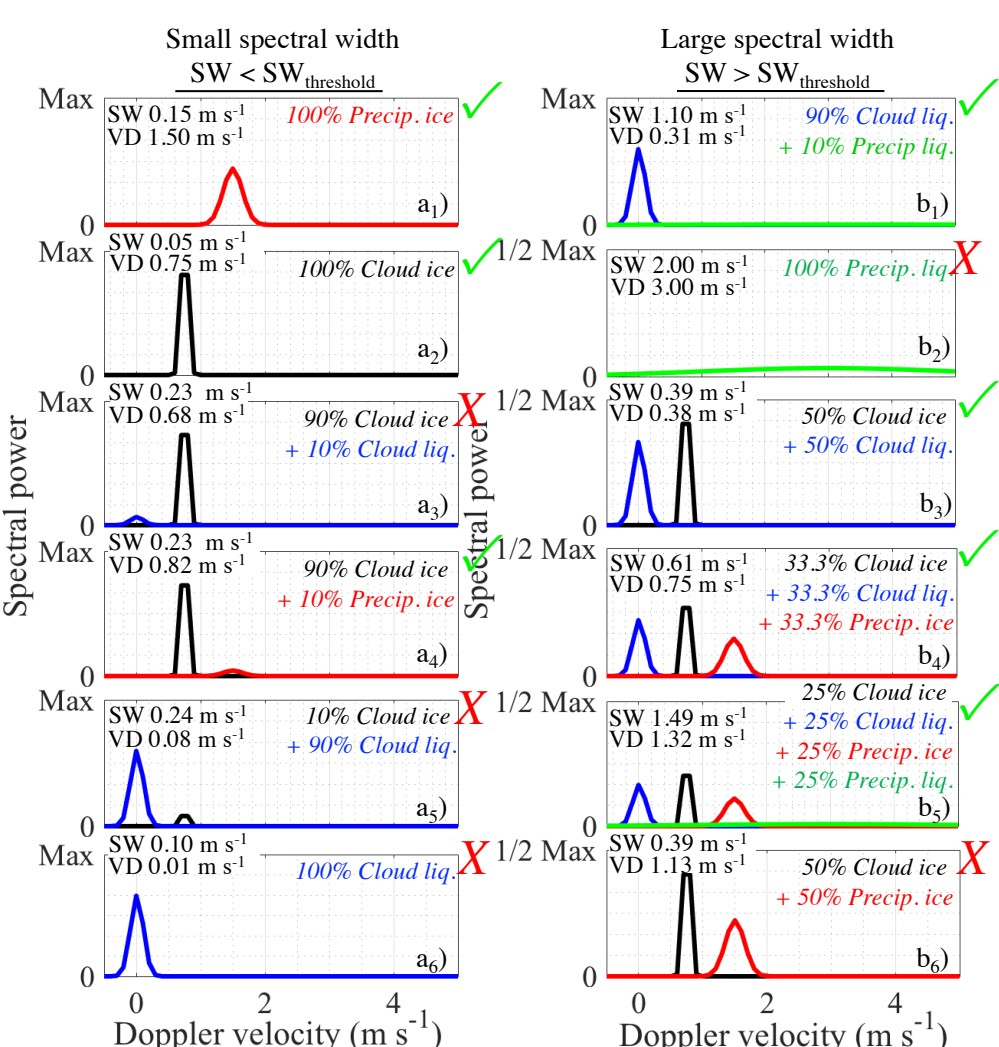

**Figure 5.** Cartoon examples of radar Doppler spectra from different hydrometeors combinations:
precipitating ice (red), cloud ice (black), precipitating water (green) and cloud water (blue). The
contribution of each hydrometeor species to the total co-polar reflectivity is indicated in the top right of
each subpanel. Each radar Doppler spectrum has been normalized to have the same total co-polar radar
reflectivity which highlights that different hydrometeor combinations generate unique mean Doppler
velocity (VD) and Doppler spectral width (SW) signatures. As discussed in Sec. 6, low spectral width
signatures are assumed to be associated with ice conditions (column a) while high spectral width signatures
are assumed to associated with liquid/mixed-phase conditions (column b). Hydrometeor combinations that
respect these assumptions are marked with √-marks. Exceptions to these rules (X-marks) are responsible
for (GO)²-SIM phase misclassifications above the level of lidar extinction. This list is not exhaustive.





## 6.1 Observational Thresholds for Hydrometeor Phase Identification

While the thresholds used for the radar reflectivity, lidar backscattered power, and lidar depolarization ratio are generally accepted by the remote sensing community, the same cannot be said about the radar Doppler velocity and Doppler spectral width thresholds suggested by Shupe (2007). The (GO)$^2$-SIM framework allows us to evaluate and/or adjust these thresholds because simulated mixing ratios of liquid and ice hydrometeors are known. The use of all such thresholds for phase classification is evaluated using joint frequency of occurrence histograms of hydrometeor mixing ratios for a single species and forward-simulated observable values (resulting from all hydrometeor types; Fig. 6). This exercise is repeated for each forward-simulation of the ensemble in order to provide a measure of uncertainty and ensure that the choice of empirical relationship does not affect our conclusions.

As one example, the joint frequency of occurrence histogram of lidar total co-polar backscattered power ($\beta_{copol,total,detect}$) and cloud liquid mixing ratio is plotted with the objective of isolating cloud ice particles from cloud water droplets (Fig. 6a$_1$, black contour lines). Two distinct clusters are evident in the joint histogram in Fig. 6a$_1$: 1) $\beta_{copol,total,detect}$ between $10^{-6.7}$ m$^{-1}$sr$^{-1}$ and $10^{-5.1}$ m$^{-1}$sr$^{-1}$ for cloud liquid water mixing ratios between $10^{-10.6}$ kg kg$^{-1}$ and $10^{-8.8}$ kg kg$^{-1}$ which we conclude result primarily from cloud ice particle contributions, and 2) $\beta_{copol,total,detect}$ between $10^{-4.6}$ m$^{-1}$sr$^{-1}$ and $10^{-3.8}$ m$^{-1}$sr$^{-1}$ for cloud liquid water mixing ratios between $10^{-6.4}$ kg kg$^{-1}$ and $10^{-4.3}$ kg kg$^{-1}$ which we conclude result primarily from cloud liquid droplet contributions. Therefore, a threshold for best distinguishing these two distinct populations should lie somewhere between $10^{-5.1}$ m$^{-1}$sr$^{-1}$ and $10^{-4.6}$ m$^{-1}$sr$^{-1}$.

To objectively determine an appropriate threshold to separate different hydrometeor populations, we start by normalizing the joint histogram of mixing ratio values for fixed ranges of observable values of interest. This normalization is done by assigning a value of 1 to the frequency of occurrence of the most frequently occurring mixing ratio value per observable range. It is then possible to evaluate the change of this most frequently occurring mixing ratio as a function of observable value. The observable value that intersects the largest change in most frequently occurring mixing ratio is then set as the threshold value.

In the example presented in Fig. 6a$_1$, the darkest grey shading is indicative of the most frequency occurring cloud liquid mixing ratio for each lidar backscattered power range. The dotted black line in Fig. 6a$_1$ connects these most frequently occurring mixing ratio values. A curved arrow points to the largest change in most frequently occurring mixing ratio as a function of $\beta_{copol,total,detect}$. A red dashed line at $10^{-4.9}$ m$^{-1}$sr$^{-1}$ indicates the lidar backscatter value that intersects this largest change in mixing ratio and represents an objective threshold value for this example forward-simulation. As mentioned earlier, this threshold is expected to change with the choice of empirical relationships used in the forward simulator. For the 576 forward-simulator realizations of this version of ModelE outputs, the interquartile range of $\beta_{copol,total,detect}$ threshold values ranged from $10^{-5}$ m$^{-1}$sr$^{-1}$ to $10^{-4.85}$ m$^{-1}$sr$^{-1}$ (red shaded vertical column).

The different panels in Fig. 6 show that similar observational patterns occur in the water mixing ratio versus lidar or radar observable histograms such that objective thresholds for hydrometeor phase classification can be determined for all of them. The second threshold determined is for the detected lidar linear depolarization ($\delta_{detect}$), once again with the goal of separating returns dominated by cloud droplets versus cloud ice particles (Fig. 6a$_2$). If we first identify the model grid cells with backscattered power above the lidar detectability threshold of $10^{-6}$ m$^{-1}$sr$^{-1}$, the threshold to distinguish between ice particles and liquid droplets is 0.36 (cyan dashed line). In the 576 forward realizations from this version of ModelE this threshold is stable at 0.36. Note that this threshold is not allowed to fall below 0.05 m s$^{-1}$.





**Figure 6.** Example of joint frequency of occurrence histograms (contours) and normalized subsets from the joint histograms (grey shading) for one (GO)$^2$-SIM forward-realization: a$_1$) $\beta_{copol,total,detect}$, a$_2$) $\delta_{detect}$, b$_1$) $SW_{copol,detect}$, and b$_2$) $Z_{copol,total,detect}$. These are used for the determination of objective water phase classifier thresholds (vertical colored dashed lines) that are set at the observational value with the largest change (see curved arrows) in most frequently occurring mixing ratio. These thresholds are not fixed but rather re-estimated for each forward-ensemble member. The widths of the color shaded vertical columns represent the interquartile range spreads generated from 576 different forward-realizations.





The third threshold determined is the radar detected co-polar spectral width ($SW_{copol,detect}$) value that
separates ice dominated from liquid/mixed-phase dominated returns (Fig. 6b$_1$). We isolate the model grid
cells with sub-zero temperatures and look for the most appropriate $SW_{copol,detect}$ threshold between 0.2 m s$^{-1}$
and 0.5 m s$^{-1}$ to isolate the ice population. For the example forward-simulation we find a threshold of 0.31
m s$^{-1}$ (green dashed line), and over all forward-realizations this threshold ranges from 0.24 m s$^{-1}$ to 0.31 m
s$^{-1}$ (green shaded vertical column).
The last threshold determined is the radar total co-polar reflectivity detected ($Z_{copol,total,detect}$) value that
separates liquid from mixed-phase dominated returns (Fig. 6b$_2$). If we isolate the model grid cells with sub-
zero temperatures, spectral widths within the liquid/mixed-phase range, and with mean Doppler velocities
smaller than 1 m s$^{-1}$, the threshold to distinguish between liquid and mixed-phase is objectively set to -23
dBZ (orange dashed line). This threshold ranges from -23.5 dBZ to -21.0 dBZ over the 576 forward
realizations obtained from this version of ModelE outputs (orange shaded vertical column).
The objectively determined thresholds optimize the performance of the hydrometeor phase classification
algorithm and are expected to generate the best (by minimizing false detection) hydrometeor phase
classifications. Results using these flexible thresholds are compared in Sec. 6.3 to results using the fixed
empirical thresholds of Shupe (2007).
**6.2 Hydrometeor Phase Map Generation**
Hydrometeor phase maps are produced for each forward realization by applying the objectively
determined flexible thresholds or fixed thresholds modified from Shupe (2007) as illustrated in Fig. 7.

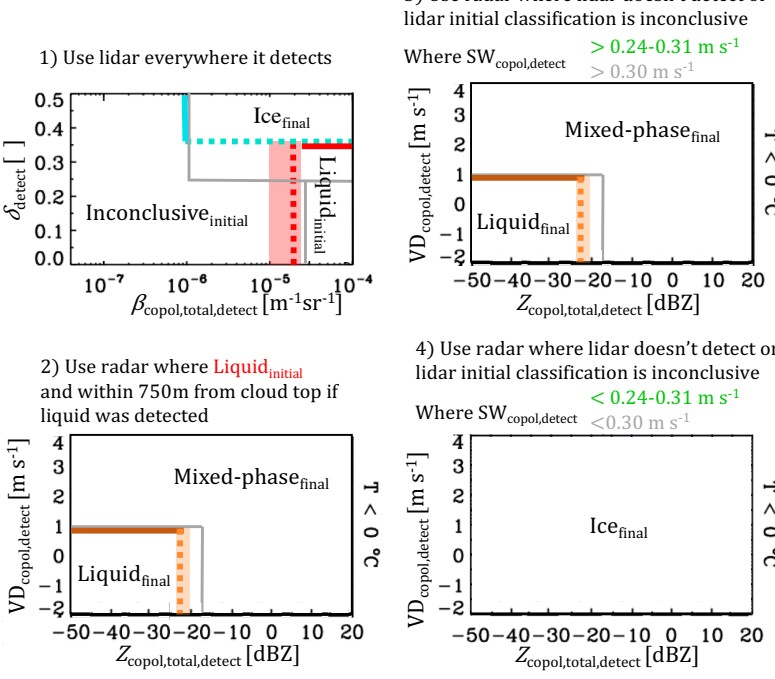

**Figure 7.** Collective illustration of hydrometeor phase classification thresholds and phase classification
sequence. Fixed thresholds modified from Shupe (2007) are displayed as grey lines. The objectively
determined flexible thresholds are displayed using dashed colored lines and colored shading as in Fig. 6.
Note that positive velocities indicate downward motion.



Thresholds are applied in sequence. Where the lidar signal is detected it is used for initial classification of
liquid-dominated grid cells (Fig. 7.1, red box) and final classification of ice-dominated grid cells (Fig. 7.1,
cyan box). Grid cells initially classified as containing liquid drops by the lidar are subsequently reclassified
as either liquid dominated  (Fig. 7.2, orange box) or mixed-phase (Fig. 7.2, outside of orange box) by the
radar which is more sensitive to the larger ice particles. Because studies suggest that supercooled water
layers extend to the tops of shallow clouds, if liquid containing grid cells were identified within 750 m of
cloud top, the radar is used to determine if there are other liquid or mixed-phase hydrometeor populations
from the range of lidar attenuation to cloud top (Fig.7.2; and just as in Shupe (2007)). Hydrometeor-
containing grid cells either not detected by the lidar or whose initial phase classification is inconclusive
(Fig. 7.1, inconclusive region) are subsequently classified using their radar moments. If radar spectral width
is above the threshold grid cells are finally classified as liquid (Fig.7.3, orange box) or mixed-phase (Fig.
7.3, outside the orange box) depending on their other radar moments. If radar spectral width is below the
threshold grid cells are finally classified as ice phase (Fig. 7.4). As a final step detected hydrometeors in
grid cells at temperatures above 0 °C are reclassified to liquid phase while those at temperatures below -40
°C are reclassified to the ice phase.
Figure 8 shows an example of (GO)$^2$-SIM water phase classification for one forward-ensemble member
using objectively determined thresholds. During the first day of this example simulation, ModelE produced
what appears to be a thick cirrus. The simulator classified this cirrus as mostly ice phase (blue). The
following day of 08/09, ModelE generated enough hydrometeors to attenuate both the forward-simulated
lidar and radar signals. The algorithm identified these hydrometeors as liquid phase (yellow). For the
following few days (08/11-08/14) deep hydrometeor systems extending from the surface to about 8 km
were produced. According to (GO)$^2$-SIM they were mostly made up of ice-phase particles (blue) with two
to three shallow mixed-phase layers at 2 km, 4 km and 7 km. Finally, on 08/14 hydrometeor systems appear
to become shallower (2-km altitudes) and liquid topped (yellow). For the entire one-year simulation, of the
333,927 detectable hydrometeor-containing grid cells, the phase classifier applied to our example forward-
simulation ensemble member identified 12.2 % pure-liquid, 68.7 % pure-ice and 19.1 % mixed-phase

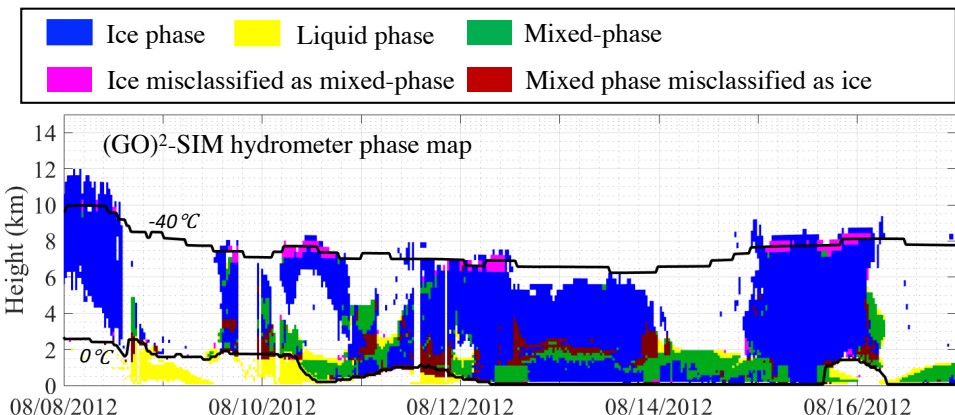

**Figure 8.** Example output from (GO)$^2$-SIM phase-classification algorithms (using objectively determined
thresholds and one set of empirical relationships in the forward-simulator). The locations of ice-phase
hydrometeors (blue), liquid-phase hydrometeors (yellow) and mixed-phase hydrometeors (green) are
illustrated. After evaluation against the original ModelE output mixing-ratios, we found that some mixed-
phase hydrometeors were misclassified as ice phase (red) and some ice-phase hydrometeors were
misclassified as mixed phase (magenta). Also indicated are the locations of the 0 °C and  -40 °C isotherms
(black lines).



conditions. Hydrometeor phase statistics estimated using this objective definition of hydrometeor phase differ by up to 60 % from those discussed at the beginning of this section that were simply based on model output nonzero mixing ratios. This indicates that a large number of grid cells containing detectable hydrometeor populations were dominated by one species and that the amounts of the other species were too small to create a phase classification signal. This highlights the need to create a framework that both objectively identifies grid cells containing detectable hydrometeors populations and determines the phase of the hydrometeors dominating them using a phase classification technique consistent with observations.

### 6.3 Phase Classification Algorithm Limitations

Hydrometeor-phase classification evaluation is facilitated in the context of forward-simulators because inputs (i.e., model-defined hydrometeor phase) are known. Model mixing-ratios are used to check for incorrect hydrometeor phase classifications over the entire forward-realization ensemble (Table 1b).

Without any ambiguity, it is possible to identify false-positive phase classifications (Table 1b). A false-positive phase classification occurs when a grid cell containing 0 kg kg$^{-1}$ of ice particles (liquid drops) is wrongly classified as ice or mixed phase (liquid or mixed phase). In this study a negligible number (0.5 %) of hydrometeor-containing model grid cells are wrongly classified as containing liquid. Similarly, a negligible number (~0.0 %) of hydrometeor-containing model grid cells are wrongly classified as containing ice particles, whereas 1.1 % of pure liquid- or ice-containing model grid cells are wrongly classified as mixed-phase. Using model mixing ratios, it is possible to determine the appropriate phase of these false-positive classifications ("False negative" row in Table 1b). An additional 1.5 % of all hydrometeor-containing model grid cells should be classified as ice phase while a negligible number (0.2 %) of liquid water is missed.

Quantifying the number of mixed-phase false negatives (i.e., the number of grid cells that should have been, but were not, classified as mixed-phase) is not as straightforward because it requires us to define mixed-phase conditions in model space. For a rough estimate of mixed-phase false negatives we check if model grid cells classified as containing a single phase contained large amounts of hydrometeors of other phase types, with large amount being defined here as a mixing-ratio greater than $10^{-5}$ kg kg$^{-1}$. This mixing-ratio amount was chosen because it is associated with noticeable changes in observables, as seen in Fig. 6. Using this mixed-phase definition, we find that 1.4 % of liquid-only classified grid cells contained large amounts of ice particles and 3.8 % of ice-only classified grid cells contained large amounts of liquid ("Questionable" row in Table 1b). Everything considered, only 6.9 % of model grid cells with detectable hydrometeor populations were misclassified according to their phase.

For completeness we examined the circumstances associated with the most frequent phase-classification errors. Most of these errors occurred above the altitude at which the lidar beam was completely attenuated, where only radar spectral widths are used to separate liquid/mixed-phase hydrometeors from ice-phase hydrometeors.

The first set of phase-classifier errors was a scarcity of pure ice particles (1.5 % false-negative ice phase). In the current (GO)$^2$-SIM implementation, ice particle populations are sometimes incorrectly classified as liquid/mixed-phase populations where cloud ice and precipitating ice hydrometeors coexist. This happens because mixtures of cloud and precipitating ice particles sometimes generate large Doppler spectral widths similar to those of mixed-phase clouds (Fig. 5b$_6$). In this example simulation ModelE produced such mixtures close to the -40 °C isotherm near the tops of deep cloud systems (e.g., Fig. 8, 08/15 around 8 km; magenta).

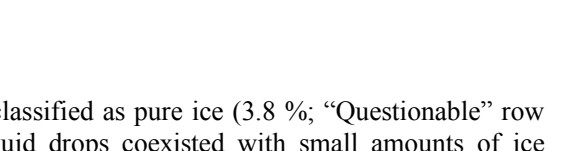

In contrast, mixed-phase conditions were sometimes misclassified as pure ice (3.8 %; "Questionable" row in Table 1b). This occurred when large amounts of liquid drops coexisted with small amounts of ice particles that generated small spectral widths incorrectly associated with pure ice particles (Fig. 5a$_5$). In this example simulation, ModelE produced such conditions just above the altitude of lidar beam extinction in cloud layers with ice falling into supercooled water layers (e.g., Fig. 8, 08/13 around 3 km; red).

Other possible misclassification scenarios associated with spectral width retrievals are presented in Fig. 5 and identified with the red X-marks. These other misclassification scenarios are not responsible for large misclassification errors here but could be in other simulations. As such, (GO)$^2$-SIM errors should be quantified every time it is applied to a new region or numerical model.

## 6.4 Sensitivity on the Choice of Threshold

The performance of the objectively determined flexible phase-classification thresholds (illustrated using colored dashed lines and shading in Fig. 7) is examined against those proposed by Shupe (2007) with one exception (illustrated using grey lines in Fig. 7). The modification to Shupe (2007) is that radar reflectivity larger than 5 dBZ are not associated with the snow category since introducing this assumption was found to increase hydrometeor-phase misclassification (not shown). From Fig. 7 it is apparent that both sets of thresholds are very similar. We estimate that hydrometeor phase frequency of occurrence produced by both threshold sets are within 6.1 % of each other and that the fixed thresholds modified from Shupe (2007) only produce phase misclassification in an additional 0.7 % of hydrometeor-containing grid cells (compare Table 1b to Table 1c). These results suggest that the use of lidar-radar threshold-based techniques for hydrometeor-phase classification depends little on the choice of thresholds.

## 7 An Ensemble Approach for Uncertainty Quantification

Owing to the limited information content in models with regard to detailed particle property information, all forward simulators must rely on a set of assumptions to estimate hydrometeor backscattered power. (GO)$^2$-SIM performs an uncertainty assessment by performing an ensemble of 576 forward simulations based on 18 different empirical relationships (relationships are listed in Table 2). The spread generated by these different combinations is expressed using median values and interquartile ranges (IQR; Table 1b,c). The fact that the largest interquartile range is 3.7 % suggests that the number of grid cells containing detectable hydrometeors as well as hydrometeor phase statistics estimated using the proposed lidar-radar algorithm are rather independent of backscattered power assumptions in the forward simulator. Nevertheless, we suggest using the full range of frequency of occurrences presented in Tables 1b,c for future model evaluation using observations.

## 8 Summary and Conclusions

Ground-based active remote sensors offer a favorable perspective for the study of shallow and multi-layer mixed-phase clouds because ground-based sensors are able to collect high resolution observations close to the surface where supercooled water layers are expected to be found. In addition, ground-based sensors have the unique capability to collect Doppler velocity information that has the potential to help identify mixed-phase conditions even in multi-layer cloud systems.

Because of differences in hydrometeor and phase definitions, among other things, observations remain incomplete benchmarks for general circulation model (GCM) evaluation. Here, a GCM-oriented ground-based observation forward-simulator [(GO)$^2$-SIM] framework for hydrometeor-phase evaluation is presented. This framework bridges the gap between observations and GCMs by mimicking observations





and their limitations and producing hydrometeor-phase maps with comparable hydrometeor definitions and
uncertainties.
Here, results over the North Slope of Alaska extracted from a one-year global ModelE (current
development version) simulation are used as an example. (GO)$^2$-SIM uses as input native resolution GCM
grid-average hydrometeor (cloud and precipitation, liquid and ice) area fractions, mixing ratios, mass-
weighted fall speeds and effective radii. These variables offer a balance between those most essential for
forward simulation of observed hydrometeor backscattering and those likely to be available from a range of
GCMs going forward, making (GO)$^2$-SIM a portable tool for model evaluation. (GO)$^2$-SIM outputs
statistics from 576 forward-simulation ensemble members all based on a different combination of eighteen
empirical relationships that relate simulated water content to hydrometeor backscattered power as would be
observed by vertically pointing micropulse lidar and Ka-band radar; The interquartile range of these
statistics being used as an uncertainty measure.
(GO)$^2$-SIM objectively determines which hydrometeor-containing model grid cells can be assessed based
on sensor capabilities, bypassing the need to arbitrarily filter trace amounts of simulated hydrometeor
mixing ratios that may be unphysical or just numerical noise. Limitations that affect sensor capabilities
represented in (GO)$^2$-SIM include attenuation and range dependent sensitivity. In this approach 78.3 % of
simulated grid cells containing nonzero hydrometeor mixing ratios were detectable and can be evaluated
using real observations, with the rest falling below the detection capability of the forward-simulated lidar
and radar leaving them unevaluated. This shows that comparing all hydrometeors produced by models with
those detected by sensors would lead to inconsistencies in the evaluation of quantities as simple as cloud
and precipitation locations and fraction.
While information can be gained from comparing the forward-simulated and observed fields, hydrometeor-
phase evaluation remains challenging owing to inconsistencies in hydrometeor-phase definitions. Models
evolve ice and liquid water species separately such that their frequency of occurrence can easily be
estimated. However, sensors record information from all hydrometeor species within a grid cell without
distinction between signals originating from ice particles or liquid drops. The additional observables of
lidar linear depolarization ratio and radar mean Doppler velocity and spectral width are forward simulated
to retrieve hydrometeor phase. The results presented here strengthen the idea that hydrometeor-phase
characteristics lead to distinct signatures in lidar and radar observables, including the radar Doppler
moments which have not been evaluated previously. Our analysis confirms that distinct patterns in
observational space are related to hydrometeor phase and an objective technique to isolate liquid, mixed-
phase and ice conditions using simulated hydrometeor mixing ratios was presented. The thresholds
produced by this technique are close to those previously estimated using real observations, further
highlighting the robustness of thresholds for hydrometeor-phase classification.
The algorithm led to hydrometeor phase misclassification in no more than 6.9 % of the hydrometeor-
containing grid cells. Its main limitations were confined above the altitude of lidar total attenuation where it
sometimes failed to identify additional mixed-phase layers dominated by liquid water drops and with few
ice particles. Using the same hydrometeor-phase definition for forward-simulated observables and real
observations should produce hydrometeor-phase statistics with comparable uncertainties. Alternatively,
disregarding how hydrometeor phase is observationally retrieved would lead to discrepancies in
hydrometeor-phase frequency of occurrence up to 40 %, a difference attributable to methodological bias
and not to model error. So, while not equivalent to model "reality" a forward-simulator framework offers
the opportunity to compare simulated and observed hydrometeor-phase maps with similar limitations and
uncertainties for a fair model evaluation.





The next steps to GCM evaluation using ground-based observations include the creation of an artifact-free observational benchmark and addressing model and observation scale differences. While the (GO)$^2$-SIM modules presented here capture sensor limitations related to backscattered power attenuations, they do not account for sensitivity inconsistencies, clutter and insect contamination, all of which affect the observations collected by the real sensors. Only thorough evaluation of observational datasets and application of masking algorithms to them can remediate these issues. Several approaches, from the subsampling of GCMs to the creation of CFADs, have been proposed to address the scale difference. A follow-up study will describe how observational resampling in the context of the cloud vertical structure approach (Rémillard and Tselioudis, 2015) can be used to account for scale differences in the context of GCM hydrometeor-phase evaluation.

(GO)$^2$-SIM is a step towards creating a fair hydrometeor-phase comparison between GCM output and ground-based observations. Owing to its simplicity and robustness, (GO)$^2$-SIM is expected to help assist in model evaluation and development for models such as ModelE, specifically with respect to hydrometeor phase in shallow cloud systems.

**Code Availability**

The ModelE code used to produce the results presented here resides within the ModelE development repository and is available upon request from the corresponding author. Results here are based on ModelE tag modelE3_2017-06-14, which is not a publicly released version of ModelE but is available on the ModelE developer repository at https://simplex.giss.nasa.gov/cgi-bin/gitweb.cgi?p=modelE.git;a=tag;h=refs/tags/modelE3_2017-06-14. The (GO)$^2$-SIM modules described in the current manuscript can be fully reproduced using the information provided. Interested parties are encouraged to contact the corresponding author for additional information on how to interface their numerical model with (GO)$^2$-SIM.

**Acknowledgements**

K. Lamer and E. Clothiaux's contributions to this research were funded by subcontract 300324 of the Pennsylvania State University with the Brookhaven National Laboratory in support to the ARM-ASR Radar Science group. The contributions of A. Fridlind, A. Ackerman, and M. Kelley were partially supported by the Office of Science (BER), U.S. Department of Energy, under agreement DE-SC0016237, the NASA Radiation Sciences Program, and the NASA Modeling, Analysis and Prediction Program. Resources supporting this work were provided by the NASA High-End Computing (HEC) Program through the NASA Center for Climate Simulation (NCCS) at Goddard Space Flight Center.

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

Contributions of clouds, surface albedos, and mixed-phase ice nucleation schemes to Arctic radiation biases
in CAM5, J. Climate, 27, 5174-5197, 2014.
Fox, N. I., and Illingworth, A. J.: The retrieval of stratocumulus cloud properties by ground-based
cloud radar, J. Appl. Meteorol., 36, 485-492, 1997.
Frey, W., Maroon, E., Pendergrass, A., and Kay, J.: Do Southern Ocean Cloud Feedbacks Matter
for 21st Century Warming?, Geophys. Res. Lett., 2017.
Gettelman, A., and Morrison, H.: Advanced two-moment bulk microphysics for global models. Part
I: Off-line tests and comparison with other schemes, J. Climate, 28, 1268-1287, 2015.
Gettelman, A., Morrison, H., Santos, S., Bogenschutz, P., and Caldwell, P.: Advanced two-moment
bulk microphysics for global models. Part II: Global model solutions and aerosol–cloud interactions, J.
Climate, 28, 1288-1307, 2015.
Hagen, M., and Yuter, S. E.: Relations between radar reflectivity, liquid-water content, and rainfall
rate during the MAP SOP, Quart. J. Roy. Meteorol. Soc.al Society, 129, 477-493, 2003.
Haynes, J., Luo, Z., Stephens, G., Marchand, R., and Bodas-Salcedo, A.: A multipurpose radar
simulation package: QuickBeam, Bull. Amer. Meteorol. Soc., 88, 1723-1727, 2007.
Heymsfield, A., Winker, D., Avery, M., Vaughan, M., Diskin, G., Deng, M., Mitev, V., and
Matthey, R.: Relationships between ice water content and volume extinction coefficient from in situ
observations for temperatures from 0 to– 86° C: Implications for spaceborne lidar retrievals, J. Appl.
Meteorol. Climatol., 53, 479-505, 2014.
Heymsfield, A. J., Winker, D., and van Zadelhoff, G. J.: Extinction-ice water content-effective
radius algorithms for CALIPSO, Geophys. Res. Lett., 32, 2005.





Hogan, R. J., Illingworth, A., O'connor, E., and Baptista, J.: Characteristics of mixed-phase clouds.
II: A climatology from ground-based lidar, Quart. J. Roy. Meteorol. Soc.al Society, 129, 2117-2134, 2003.
Hogan, R. J., Behera, M. D., O'Connor, E. J., and Illingworth, A. J.: Estimate of the global
distribution of stratiform supercooled liquid water clouds using the LITE lidar, Geophys. Res. Lett., 31,
849  2004.

Hogan, R. J., and O'Connor, E.: Facilitating cloud radar and lidar algorithms: The Cloudnet
Instrument Synergy/Target Categorization product, Cloudnet documentation, 2004.
Hogan, R. J., Mittermaier, M. P., and Illingworth, A. J.: The retrieval of ice water content from
radar reflectivity factor and temperature and its use in evaluating a mesoscale model, J. Appl. Meteorol.
Climatol., 45, 301-317, 2006.
Hu, Y., Vaughan, M., Liu, Z., Lin, B., Yang, P., Flittner, D., Hunt, B., Kuehn, R., Huang, J., and
Wu, D.: The depolarization-attenuated backscatter relation: CALIPSO lidar measurements vs. theory,
Optics Express, 15, 5327-5332, 2007a.
Hu, Y., Vaughan, M., McClain, C., Behrenfeld, M., Maring, H., Anderson, D., Sun-Mack, S.,
Flittner, D., Huang, J., Wielicki, B., Minnis, P., Weimer, C., Trepte, C., and Kuehn, R.: Global statistics of
liquid water content and effective number concentration  of water clouds over ocean derived from
combined CALIPSO and MODIS  measurements, Atmos. Chem. Phys., 7, 3353--3359, 10.5194/acp-7-
3353-2007, 2007b.
Hu, Y., Winker, D., Vaughan, M., Lin, B., Omar, A., Trepte, C., Flittner, D., Yang, P., Nasiri, S. L.,
and Baum, B.: CALIPSO/CALIOP cloud phase discrimination algorithm, J. Atmos. Ocean. Technol., 26,
865  2293-2309, 2009.

Hu, Y., Rodier, S., Xu, K. m., Sun, W., Huang, J., Lin, B., Zhai, P., and Josset, D.: Occurrence,
liquid water content, and fraction of supercooled water clouds from combined CALIOP/IIR/MODIS
measurements, J. Geophys. Res.: Atmos., 115, 2010.
Huang, Y., Siems, S. T., Manton, M. J., Hande, L. B., and Haynes, J. M.: The structure of low-
altitude clouds over the Southern Ocean as seen by CloudSat, J. Climate, 25, 2535-2546, 2012a.
Huang, Y., Siems, S. T., Manton, M. J., Protat, A., and Delanoë, J.: A study on the low-altitude
clouds over the Southern Ocean using the DARDAR-MASK, J. Geophys. Res.: Atmos., 117, 2012b.
Intrieri, J., Shupe, M., Uttal, T., and McCarty, B.: An annual cycle of Arctic cloud characteristics
observed by radar and lidar at SHEBA, J. Geophys. Res.: Oceans, 107, 2002.
Kalesse, H., Szyrmer, W., Kneifel, S., Kollias, P., and Luke, E.: Fingerprints of a riming event on
cloud radar Doppler spectra: observations and modeling, Atmos. Chem. Phys., 16, 2997-3012, 2016.
Kay, J. E., Bourdages, L., Miller, N. B., Morrison, A., Yettella, V., Chepfer, H., and Eaton, B.:
Evaluating and improving cloud phase in the Community Atmosphere Model version 5 using spaceborne
lidar observations, J. Geophys. Res.: Atmos., 121, 4162-4176, 2016.
Kikuchi, K., Tsuboya, S., Sato, N., Asuma, Y., Takeda, T., and Fujiyoshi, Y.: Observation of
wintertime clouds and precipitation in the Arctic Canada (POLEX-North), J. Meteorol. Soc. Japan. Ser. II,
882  60, 1215-1226, 1982.



Klein, S. A., McCoy, R. B., Morrison, H., Ackerman, A. S., Avramov, A., Boer, G. d., Chen, M.,
Cole, J. N., Del Genio, A. D., and Falk, M.: Intercomparison of model simulations of mixed-phase clouds
observed during the ARM Mixed-Phase Arctic Cloud Experiment. I: Single-layer cloud, Quart. J. Roy.
Meteorol. Soc., 135, 979-1002, 2009.
Kollias, P., Miller, M. A., Luke, E. P., Johnson, K. L., Clothiaux, E. E., Moran, K. P., Widener, K.
B., and Albrecht, B. A.: The Atmospheric Radiation Measurement Program cloud profiling radars: Second-
generation sampling strategies, processing, and cloud data products, J. Atmos. Ocean. Technol., 24, 1199-
890 1214, 2007.

Kollias, P., Rémillard, J., Luke, E., and Szyrmer, W.: Cloud radar Doppler spectra in drizzling
stratiform clouds: 1. Forward modeling and remote sensing applications, J. Geophys. Res.: Atmos., 116,
893 2011.

Kollias, P., Clothiaux, E. E., Ackerman, T. P., Albrecht, B. A., Widener, K. B., Moran, K. P., Luke,
E. P., Johnson, K. L., Bharadwaj, N., and Mead, J. B.: Development and applications of ARM millimeter-
wavelength cloud radars, Meteorological Monographs, 57, 17.11-17.19, 2016.
Kuehn, R., Holz, R., Eloranta, E., Vaughan, M., and Hair, J.: Developing a Climatology of Cirrus
Lidar Ratios Using Univeristy of Wisconsin HSRL Observations, EPJ Web of Conferences, 2016, 16009,
Liao, L., and Sassen, K.: Investigation of relationships between Ka-band radar reflectivity and ice
and liquid water contents, Atmospheric Res., 34, 231-248, 1994.
Liu, C.-L., and Illingworth, A. J.: Toward more accurate retrievals of ice water content from radar
measurements of clouds, J. Appl. Meteorol., 39, 1130-1146, 2000.
McCoy, D. T., Tan, I., Hartmann, D. L., Zelinka, M. D., and Storelvmo, T.: On the relationships
among cloud cover, mixed-phase partitioning, and planetary albedo in GCMs, J. Advances in Modeling
Earth Systems, 8, 650-668, 2016.
O'Connor, E. J., Illingworth, A. J., and Hogan, R. J.: A technique for autocalibration of cloud lidar,
J. Atmos. Ocean. Technol., 21, 777-786, 2004.
Rémillard, J., and Tselioudis, G.: Cloud regime variability over the Azores and its application to
climate model evaluation, J. Climate, 28, 9707-9720, 2015.
Sassen, K.: Ice cloud content from radar reflectivity, J. climate and Appl. Meteorol., 26, 1050-1053,
911 1987.

Sassen, K.: The polarization lidar technique for cloud research: A review and current assessment,
Bull. Amer. Meteorol. Soc., 72, 1848-1866, 1991.
Sato, N., Kikuchi, K., Barnard, S. C., and Hogan, A. W.: Some characteristic properties of ice
crystal precipitation in the summer season at South Pole Station, Antarctica, J. Meteorol. Soc. Japan. Ser.
II, 59, 772-780, 1981.
Sauvageot, H., and Omar, J.: Radar reflectivity of cumulus clouds, J. Atmos. Ocean. Technol., 4,
918 264-272, 1987.





Schmidt, G. A., Kelley, M., Nazarenko, L., Ruedy, R., Russell, G. L., Aleinov, I., Bauer, M., Bauer,
S. E., Bhat, M. K., and Bleck, R.: Configuration and assessment of the GISS ModelE2 contributions to the
CMIP5 archive, J. Adv. Model. Earth Syst., 6, 141-184, 2014.
Sekhon, R. S., and Srivastava, R.: Doppler radar observations of drop-size distributions in a
thunderstorm, J. Atmos. Sci., 28, 983-994, 1971.
Shupe, M. D.: A ground-based multisensor cloud phase classifier, Geophys. Res. Lett., 34, 2007.
Tan, I., and Storelvmo, T.: Sensitivity study on the influence of cloud microphysical parameters on
mixed-phase cloud thermodynamic phase partitioning in CAM5, J. Atmos. Sci., 73, 709-728, 2016.
Tan, I., Storelvmo, T., and Zelinka, M. D.: Observational constraints on mixed-phase clouds imply
higher climate sensitivity, Science, 352, 224-227, 2016.
Tatarevic, A., and Kollias, P.: User's Guide to Cloud Resolving Model Radar Simulator (CR-SIM),
McGill University Clouds Research Group, Document available at http://radarscience.weebly.com/radar-
simulators.html. 2015.
Yoshida, R., Okamoto, H., Hagihara, Y., and Ishimoto, H.: Global analysis of cloud phase and ice
crystal orientation from Cloud-Aerosol Lidar and Infrared Pathfinder Satellite Observation (CALIPSO)
data using attenuated backscattering and depolarization ratio, J. Geophys. Res.: Atmos., 115, 2010.
Zhang, Y., Xie, S., Klein, S. A., Marchand, R., Kollias, P., Clothiaux, E. E., Lin, W., Johnson, K.,
Swales, D., and Bodas-Salcedo, A.: The ARM Cloud Radar Simulator for Global Climate Models: A New
Tool for Bridging Field Data and Climate Models, Bull. Amer. Meteorol. Soc., 2017.