# Peer review of "(GO)2-SIM: A GCM-Oriented Ground-Observation Forward-Simulator Framework for Objective Evaluation of Cloud and Precipitation Phase"

_Geoscientific Model Development, 2018_

## Short Comment (SC1) · 22 Jun 2018

As outlined in https://www.geoscientific-model-development.net/about/manuscript_types.html GMD is expecting that the model code is publicly available through a permanent arrangement. Given the impermanence of email addresses, GMD encourages authors acting as a point of contact for obtaining the code to improve the availability with a more permanent and public arrangement. When copyright or licensing restrictions prevent the public release of model code, or in the cases where there is some other good reason for not allowing public access to the code, authors need to state the

reasons for why access is restricted and need to explain how access can be obtained (e.g. signing a license agree or join a consortium).

Lutz Gross GMD Executive Editor

---

## Referee Comment (RC1) · Anonymous Referee #1 · 23 Jul 2018

This manuscript describes a forward-simulator designed to take general circulation model (GCM) hydrometeor fields and provide lidar and Ka-band radar measurements. Instead of basing the forward calculations on fundamental radiative transfer theory, the calculations largely rely on 18 empirical relationships to convert the hydrometeor fields into lidar and radar measurements. The authors interpret the ensemble of different empirical relationships as a measure of uncertainty. Phase classification is based on the forward-model calculations.

The article is well written and the descriptions of forward-model calculations like this

are a useful contribution to the peer-reviewed literature. My criticisms are mostly philosophical in nature.

1. The advantage of basing forward calculations on empirical relationships, as opposed to fundamental radiative transfer and scattering theory, is not well established in the manuscript. One justification for the approach in the paper is that using empirical relationships means that one does not need to make assumptions about scatterers (e.g., a spherical assumption), but this simply exchanges a known assumption with an assumption (or set of assumptions) hidden in the empirical relationships. And most of these empirical relationships are actually retrievals, just inverted! If we're going to do forward calculations based on retrievals, we might as well just use the retrievals on the observations and cast all the quantities in terms of geophysical variables, which are easier to interpret. This approach seems like a big step back compared to performing fundamental radiative transfer/scattering calculations on the model fields, which yields an independent forward calculation of the observational fields. Furthermore, the assumptions in the empirical relationships may not be consistent with the assumptions in the model cloud microphysical parameterization (e.g., the assumed distributions). Consistent forward calculations of model variables should use assumptions consistent with the cloud-physics scheme in the model.

2. The manuscript advocates a phase determination that is solely in forward-calculation space and fairly well articulates the reason for this. However, this approach does not take advantage of knowing the actual hydrometeor fields, and therefore this discards a great deal of potentially useful information. Is there any way the approach in the manuscript can take some advantage of the fields in hydrometeor (model) space?

3. Constructing an ensemble of forward calculations based on different empirical relationships is a good idea, but it is a stretch to portray it as quantifying uncertainty. The authors have no way to know to what extent the results from these calculations actually map to the PDF of possible outcomes. It is useful but is not statistically defensible to call it UQ. The authors should much more carefully word this claim.

[Figure]

4. The calculations are based on 30-minute instantaneous model hydrometeor fields. The article is focused on the actual forward calculations of the microphysical fields, but comparison of forward-model calculations and observations necessarily includes assumptions of spatial and temporal scale. Would the authors please discuss with a bit more detail on how the forward calculations (30-minute instantaneous calculations of lidar and radar fields) would be compared to observations? If nothing else, this would provide some guidance for readers using their forward simulator.

Please also note the supplement to this comment:
https://www.geosci-model-dev-discuss.net/gmd-2018-99/gmd-2018-99-RC1-supplement.pdf

---

## Referee Comment (RC2) · Anonymous Referee #2 · 25 Jul 2018

The manuscript describes a forward simulator to convert hydrometer fields in lidar and radar properties. The approach to convert modeled hydrometer fields in lidar and radar measurements (also taking measurement restrictions into account) is very valuable. The paper is well written and suitable for publication after addressing the following points:

Although it is frequently stressed in the manuscript that the radar is very sensible to particle size none of the empirical equations takes the particle size into account.

[Figure]

The motivation for the ice lidar ratio of 25.7 sr (Eq. 7) is not motivated. Additionally the lidar ratio is often dependent on the particle size which is not addressed in the manuscript. No multiple scattering is simulated even for water clouds or thick ice clouds.

Please give a reference for radar attenuation (Eq. 24b).

The meaning of the terms in Eq. 29 is not completely clear to me. Please give the derivation of Eq. 29.

A number of empirical equations are used to estimate the uncertainties. Although each formula is valuable for specific situations I am not sure if their ensemble covers the whole range of variability of ModelE output. A forward model using the modelled effective radius might help.

---

## Author Comment (AC1) · 6 Sep 2018

The authors would like to thank the reviewer for their insightful comments. A point by point response to the reviewer's comments, along with changes made to the manuscript as a result, are included below.

R1. The advantage of basing forward calculations on empirical relationships, as opposed to fundamental radiative transfer and scattering theory, is not well established in the manuscript. One justification for the approach in the paper is that using empirical

relationships means that one does not need to make assumptions about scatterers (e.g., a spherical assumption), but this simply exchanges a known assumption with an assumption (or set of assumptions) hidden in the empirical relationships. And most of these empirical relationships are actually retrievals, just inverted! If we're going to do forward calculations based on retrievals, we might as well just use the retrievals on the observations and cast all the quantities in terms of geophysical variables, which are easier to interpret. This approach seems like a big step back compared to performing fundamental radiative transfer/scattering calculations on the model fields, which yields an independent forward calculation of the observational fields. Furthermore, the assumptions in the empirical relationships may not be consistent with the assumptions in the model cloud microphysical parameterization (e.g., the assumed distributions). Consistent forward calculations of model variables should use assumptions consistent with the cloud-physics scheme in the model.

A1. In response to this comment by the reviewer we now elaborate within the manuscript on the reasoning behind our approach:

"Hydrometeor properties that impact backscattering include size, phase, composition, geometrical shape, orientation and bulk density. Were plausible representations for these hydrometeor properties available as part of the model formulation, fundamental radiative scattering transfer calculations would be the most accurate way to transform model hydrometeor properties to observables. However, in most GCMs such detailed hydrometeor information is highly simplified (e.g., fixed particle size distribution shapes) or not explicitly represented (e.g., orientation and realistic geometrical shape), complicating the process of performing direct radiative scattering transfer calculations. Chepfer et al. (2008) proposed an approach by which lidar backscattered power can be forward-simulated using model output hydrometeor effective radius. Their approach, based on Mie theory, relies on the assumption that cloud particles (both liquid and ice) are spherical and requires additional assumptions about hydrometeor size distributions and scattering efficiencies. Similarly, the COSP (Bodas-Salcedo et al., 2011) and

ARM Cloud Radar Simulator for GCMs (Zhang et al., 2017) packages both use Quick-Beam for the estimation of radar backscattered power (i.e., radar reflectivity; Haynes et al., 2007). QuickBeam computes radar reflectivity using Mie theory again under the assumption that all hydrometeor species are spherical and by making additional assumptions about the shape of hydrometeor size distributions as well as mass-size and diameter-density relationships. While some of these assumptions may be consistent with the assumptions in model cloud microphysical parameterizations, some are not adequately realistic (e.g., spherical ice) or complete for accurate backscattering estimation and it is typically very difficult to establish the sensitivity of results to all such assumptions.

To avoid having to make ad hoc assumptions about hydrometeor shapes, orientations, and compositions, which are properties that also remain poorly documented in nature, (GO)2-SIM employs empirical relationships to convert model output to observables. These empirical relationships based on observations, direct or retrieved with their own sets of underlying assumptions, are expected to capture at least part of the natural variability in hydrometeor properties. Additionally empirical relationships are computationally less expensive to implement than direct radiative scattering calculations, thus enabling the estimation of an ensemble of backscattering calculations using a range of assumptions in an effort to quantify part of the backscattering uncertainty (see Sec. 7). The empirical relationships proposed require few model inputs, potentially enhancing consistency in applying (GO)2-SIM to models with differing microphysics scheme assumptions and complexity. Section 6 will show that, while the empirical relationships employed in (GO)2-SIM may not be as exact as direct radiative scattering calculations, they produce backscattering estimates of sufficient accuracy for hydrometeor phase classification, which is the main purpose of (GO)2-SIM at this time."

R2. The manuscript advocates a phase determination that is solely in forward-calculation space and fairly well articulates the reason for this. However, this approach does not take advantage of knowing the actual hydrometeor fields, and therefore this
discards a great deal of potentially useful information. Is there any way the approach in the manuscript can take some advantage of the fields in hydrometeor (model) space?

A2. As articulated in the manuscript our goal is "[. . .] development of a phase classification algorithm that can be applied to observables, forward-simulated and real." This explains why we avoided developing a hydrometeor-phase classifier dependent on model output quantities that are not accessible via observations. Rather, we take advantage of the fields in model space by using them to 1) evaluate the ability of Doppler velocity and Doppler spectral width observations to be used for hydrometeor phase classification (a concept which was developed empirically and was not formally evaluated) and to 2) select optimum classification thresholds to minimize false detection in model space.

This reasoning is expressed in the following modified manuscript excerpts:

"While the thresholds used for the radar reflectivity, lidar backscattered power, and lidar depolarization ratio are generally accepted by the remote sensing community, the same cannot be said about the radar Doppler velocity and Doppler spectral width thresholds suggested by Shupe (2007). Because simulated mixing ratios of liquid and ice hydrometeors are known in the (GO)2-SIM framework, the use and choice of all such thresholds for phase classification can be evaluated using joint frequency of occurrence histograms of hydrometeor mixing ratios for a single species and forward-simulated observable values (resulting from all hydrometeor types; Fig. 6)."

"The objectively determined thresholds, based on model output mixing ratios, optimize the performance of the hydrometeor phase classification algorithm and are expected to generate the best (by minimizing false detection) hydrometeor phase classifications. Results using these objective flexible thresholds are compared in Sec. 6.4 to results using the fixed empirical thresholds of Shupe (2007)."

"The performance of the objectively determined flexible phase-classification thresholds (illustrated using colored dashed lines and shading in Fig. 7) is examined against

those empirically derived by Shupe (2007) with one exception (illustrated using grey lines in Fig. 7). The modification to Shupe (2007) is that radar reflectivity larger than 5 dBZ are not associated with the snow category since introducing this assumption was found to increase hydrometeor-phase misclassification (not shown). From Fig. 7 it is apparent that both sets of thresholds are very similar. We estimate that hydrometeor phase frequency of occurrence produced by both threshold sets are within 6.1 % of each other and that the fixed empirical thresholds modified from Shupe (2007) only produce phase misclassification in an additional 0.7 % of hydrometeor-containing grid cells (compare Table 1b to Table 1c). These results suggest that the use of lidar-radar threshold-based techniques for hydrometeor-phase classification depends little on the choice of thresholds."

R3. Constructing an ensemble of forward calculations based on different empirical relationships is a good idea, but it is a stretch to portray it as quantifying uncertainty. The authors have no way to know to what extent the results from these calculations actually map to the PDF of possible outcomes. It is useful but is not statistically defensible to call it UQ. The authors should much more carefully word this claim.

A3. The authors agree with the reviewer that the 576 forward-simulations performed do not cover the entire range of possible scattering assumptions. The following manuscript changes reflect this reality:

"Additionally empirical relationships are computationally less expensive to implement than direct radiative scattering calculations, thus enabling the estimation of an ensemble of backscattering calculations using a range of assumptions in an effort to quantify part of the backscattering uncertainty (see Sec. 7)."

"(GO)2-SIM performs an uncertainty assessment by performing an ensemble of 576 forward simulations based on 18 different empirical relationships (relationships are listed in Table 2). While the relationships used do not cover the entire range of possible backscattering assumptions, they represent an attempt at uncertainty quantification

and illustrate a framework for doing so. [. . .] Nevertheless, we suggest using the full range of frequency of occurrences presented in Tables 1b,c for future model evaluation using observations and acknowledge that additional uncertainty is most likely present."

R4. The calculations are based on 30-minute instantaneous model hydrometeor fields. The article is focused on the actual forward calculations of the microphysical fields, but comparison of forward-model calculations and observations necessarily includes assumptions of spatial and temporal scale. Would the authors please discuss with a bit more detail on how the forward calculations (30-minute instantaneous calculations of lidar and radar fields) would be compared to observations? If nothing else, this would provide some guidance for readers using their forward simulator.

A4. We now elaborate more on this topic and provide an updated flow chart: "A follow-up study will describe an approach by which vertical and temporal resampling of observations can help reduce the scale gap. Furthermore, it will be showed that, using simplified model evaluation targets based on three atmospheric regions separated by constant pressure levels, ground-based observations can be used for GCM hydrometeor-phase evaluation."

Figure 1. (GO)2-SIM framework. (GO)2-SIM emulates two types of remote sensors: Ka-band Doppler radars (dark gray shading) and 532 nm polarimetric lidars (light gray shading). It then tunes and applies a common phase-classification algorithm (white boxes) to both observed (upper section) and forward-simulated (bottom section) fields. Follow-on work will describe how observation can be post-processed and resampled to reduce the scale gap before model evaluation can be performed.

Please also note the supplement to this comment:
https://www.geosci-model-dev-discuss.net/gmd-2018-99/gmd-2018-99-AC1-supplement.pdf
* * *
**Observation post-processing**

Ground-based zenith measurements

Radar    Lidar    Sonde  MWR
kazrge .b1  mplcmask.c1  wnpn.b1  los.b1

Data quality control

LWP

T
P

$P_{co+cross,\ corr}$

$\delta$

Calibration

$\beta_{co+cross}$

$\delta$

Isolate obs. from noise

$\beta_{copol,detect}$

$\delta_{detect}$

$SNR_{copol}$
$Z_{copol}$
$VD_{copol}$
$SW_{copol}$

Isolate meaningful obs. from noise, insects and clutter

$Z_{copol,\ detect}$
$VD_{copol,\ detect}$
$SW_{copol,\ detect}$

Temporal and Vertical Resampling

Addressing the scale gap

$LWP^R$
$P^R$
$T^R$
$\beta^R_{copol,\ detect}$
$\delta^R_{detect}$
$Z^R_{copol,\ detect}$
$VD^R_{copol,\ detect}$
$SW^R_{copol,\ detect}$

Hydrometeor phase classifier

Hydrometeor phase map

GCM output - ModelE

Re

$T$
$P$
$q$
$f$

$V$

Lidar backscatter

$\beta_{copol}$

Partial and total attenuation

$\beta_{copol\ detect}$

Lidar linear depol. ratio

$\delta_{detect}$

Radar backscatter

$Z_{copol}$

Attenuation Range sensitivity

$Z_{copol\ detect}$

Doppler observables

$VD_{copol\ detect}$
$SW_{copol\ detect}$

$(GO)^2$-SIM

LWP
T
P
$\beta_{copol,\ detect}$
$\delta_{detect}$
$Z_{copol,\ detect}$
$VD_{copol,\ detect}$
$SW_{copol,\ detect}$

Hydrometeor phase map

*Sec. 2.*   *Sec. 3.*          *Sec. 4.*          *Sec. 5.*          *Sec. 6.*

**Fig. 1.** Described in A4

---

## Author Comment (AC2) · 6 Sep 2018

The authors would like to thank the reviewer for their insightful comments. A point by point response to the reviewer's comments, along with changes made to the manuscript as a result, are included below. (Please see the pdf document attached at the end of this document for a better rendering of the mathematical equations)

R1. Although it is frequently stressed in the manuscript that the radar is very sensible to particle size none of the empirical equations takes the particle size into account.

[Figure]

A1. The following manuscript changes have been made to address the reviewer's comment:

"(GO)2-SIM relies on water content-based empirical relationships to estimate cloud liquid water (cl), cloud ice (ci), precipitating liquid water (pl) and precipitating ice (pi) radar reflectivity. Different relationships are used for each species to account for the fact that hydrometeor mass and size both affect radar reflectivity."

"Figure 3b illustrates the fact that for all these empirical relationships increasing water content leads to increasing radar reflectivity. As already mentioned, radar reflectivity is approximately related to the sixth power of the particle size, which explains why, for the same water content, precipitating hydrometeors are associated with greater reflectivity than cloud hydrometeors."

R2. The motivation for the ice lidar ratio of 25.7 sr (Eq. 7) is not motivated. Additionally the lidar ratio is often dependent on the particle size which is not addressed in the manuscript.

A2. The following manuscript changes have been made to address the reviewer's comment:

"Lidar co-polar backscattered power ($\beta$_(copol,species) [mˆ(-1) srˆ(-1)]) generated by each hydrometeor species is related to lidar extinction ãĂŰ($\sigma$ãĂŮ_(copol,species) [m-1]) through the lidar ratio (Sspecies [sr]):

$\beta$_(copol,cl)= ((1)/(S_cl) $\sigma$_(copol,cl) ). (6) $\beta$_(copol,ci)= ((1)/(S_ci) $\sigma$_(copol,ci) ). (7)

While constant values are used for the lidar ratios of liquid and ice clouds in this version of the forward-simulator, we acknowledge that in reality they depend on particle size. O'Connor et al. (2004) suggest that a liquid cloud lidar ratio (Scl) of 18.6 sr is valid for cloud liquid droplets smaller than 25 $\mu$m, which encompasses the median diameter expected in the stratiform clouds simulated here. Kuehn et al. (2016) observed layer-averaged lidar ratios in ice clouds (Sci) ranging from 15.1 to 36.3 sr. Sensitivity tests

indicate that adjusting the ice cloud lidar ratio to either of these extreme values in the forward-simulator increases the number of detectable hydrometeors by no more than 0.6 %, changes the hydrometeor phase frequency of occurrence statistics by less than 0.4% and causes less than a 0.1% change in phase-classification errors (not shown). Given these results, the ice cloud lidar ratio is set to the constant value of 25.7 sr, which corresponds to the mean value observed by Kuehn et al. (2016)"

R3. No multiple scattering is simulated even for water clouds or thick ice clouds.

A3. The following manuscript changes have been made to address the reviewer's comment:

"Lidar attenuation is exponential and two-way as it affects the lidar power on its way out and back:

$$\beta_{(copol,total,att)} = \beta_{(copol,total)} \, e^{-2\eta\tau}. \quad (22)$$

Note that in some instances multiple scattering occurs before the lidar signal returns to the sensor, thus amplifying the returned signal. In theory, the multiple scattering coefficient ($\eta$) varies from 0 to 1. Sensors with large fields of view, such as satellite-based lidars, are more likely to be impacted by multiple scattering than others (Winker, 2003). In the current study, for which a ground-based lidar is simulated, a multiple scattering coefficient of unity is used. A sensitivity test in which this coefficient was varied from 0.7, such as that implemented in the CALIPSO satellite lidar simulator of Chepfer et al. (2008), to 0.3, representing an extreme case, indicated that multiple scattering had a negligible impact (less than 1%) on the number of hydrometeors detected, the hydrometeor phase frequency of occurrence statistics, and in phase classification error (not shown)."

"According to an analysis of CALIPSO observations by Cesana and Chepfer (2013), cloud ice particle cross-polar backscattering ($\beta_{(crosspol,ci,detect)}$ [m^(-1) sr^(-1) ]) and cloud liquid droplet cross-polar backscattering ($\beta_{(crosspol,cl,detect)}$ [m^(-1) sr^(-

1) ]) can be approximated using the following relationships:

ãĂŰ $\beta$ãĂŮ_(crosspol,ci,detect)=0.29ãĂŰ ($\beta$ãĂŮ_(copol,ci,detect )+$\beta$_(crosspol,ci,detect)), (26b)

ãĂŰ $\beta$ãĂŮ_(crosspol,cl,detect)= 1.39ãĂŰ ($\beta$ãĂŮ_(copol,cl,detect)+$\beta$_(crosspol,cl,detect)) +1.76 ãĂŰ10ãĂŮˆ(-2 ) ãĂŰ ($\beta$ãĂŮ_(copol,cl,detect)+$\beta$_(crosspol,cl,detect)) $\approx$0. (26c)

For reasons mentioned in Sec. 4.1, multiple scattering is considered negligible in the current study such that cloud-liquid droplet cross-polar backscattering is assumed to be zero under all conditions."

R4. Please give a reference for radar attenuation (Eq. 24b).

A4. The manuscript was modified to include a reference to Ellis, S. M., and Vivekanandan, J.: Liquid water content estimates using simultaneous S and Ka band radar measurements, Radio Science, 46, 2011:

"At 8.56 mm (Ka-band) total co-polar attenuated reflectivity (Z_(copol,total,att) [dBZ]) is given by:

Z_(copol,total,att)=Z_(copol,total)-2$\int$ _(z = 0)$\vartheta[a(WC\_pl + WC\_cl)]dh$, (24)

where attenuation is controlled by the wavelength-dependent attenuation coefficient a ([dB km-1 (g m-3)-1]) which we take to be 0.6 at Ka-band (Ellis and Vivekanandan, 2011), by the water contents of cloud liquid (WCcl [gãĂŰ mãĂŮˆ(-3) ]) and precipitating liquid (WCcl [gãĂŰ mãĂŮˆ(-3) ] ), and by the thickness of the liquid layer."

R5. The meaning of the terms in Eq. 29 is not completely clear to me. Please give the derivation of Eq. 29.

A5. A reference to Everitt, B., and Hand, D.: Mixtures of normal distributions, in: Finite Mixture Distributions, Springer, 25-57, 1981 was added. A derivation of the first five central moments of a two-component univariate normal mixture is presented in their book. The following manuscript changes were made to improve clarity: "Total mean

Doppler velocity detected (VDcopol,detect [m sˆ(-1) ]) is the reflectivity-weighted sum of the mass-weighted fall velocity of each hydrometeor species (Vspecies[m sˆ(-1) ]):

$$\text{VD\_(copol,detect)}= \sum\_(species=cl,pl,ci,pi)P\_species V\_species, (28)$$

where the mass-weighted fall velocity of each hydrometeor species (Vspecies[m sˆ(-1) ]) is a model output. Total Doppler spectral width (SWcopol,detect [m sˆ(-1) ]) is more complex and can be estimated following a statistical method similar to that described by Everitt and Hand (1981). It takes into consideration the properties of each individual hydrometeor species through their respective fall speed (Vspeies [m sˆ(-1) ]) and spectral width (SWspecies [m sˆ(-1) ]) in relation to the properties of the hydrometeor population as a whole through the total mean Doppler velocity detected (VD\_(copol,detect)) estimated in Eq. 28:

$$\text{SW\_(copol,detect)}= \sum\_(species=cl,pl,ci,pi)P\_species(SW\_species2 + (V\_species - VD\_(copol,detect))2), (29)$$

where the spectral widths of individual species (SWspecies) are assigned climatological values. These climatological values are SW_cl=0.10 m sˆ(-1), SW_ci=0.05 m sˆ(-1), SW_pi=0.15 m sˆ(-1) and SW_pl=2.00 m sˆ(-1) (Kalesse et al., 2016)."

R6. A number of empirical equations are used to estimate the uncertainties. Although each formula is valuable for specific situations I am not sure if their ensemble covers the whole range of variability of ModelE output. A forward model using the modelled effective radius might help.

A6. The authors agree with the reviewer that the 576 forward-simulations performed do not cover the entire range of possible scattering assumptions. The following manuscript changes reflect this reality:

"Additionally empirical relationships are computationally less expensive to implement than direct radiative scattering calculations, thus enabling the estimation of an ensemble of backscattering calculations using a range of assumptions in an effort to quantify part of the backscattering uncertainty (see Sec. 7)."

(GO)2-SIM performs an uncertainty assessment by performing an ensemble of 576 forward simulations based on 18 different empirical relationships (relationships are listed in Table 2). While the relationships used do not cover the entire range of possible backscattering assumptions, they represent an attempt at uncertainty quantification and illustrate a framework for doing so. [. . .] Nevertheless, we suggest using the full range of frequency of occurrences presented in Tables 1b,c for future model evaluation using observations and acknowledge that additional uncertainty is most likely present."

Please also note the supplement to this comment:
https://www.geosci-model-dev-discuss.net/gmd-2018-99/gmd-2018-99-AC2-supplement.pdf
* * *

---

## Author Comment (AC3) · 6 Sep 2018

In response to this comment the authors have modified the code availability section as follows:

"Results here are based on ModelE tag modelE3_2017-06-14, which is not a publicly released version of ModelE but is available on the ModelE developer repository at https://simplex.giss.nasa.gov/cgi-bin/gitweb.cgi?p=modelE.git;a=tag;h=refs/tags/modelE3_2017-06-14. The (GO)2-SIM

[Figure]

modules described in the current manuscript can be fully reproduced using the information provided. Interested parties are encouraged to contact the corresponding author for additional information on how to interface their numerical model with (GO)2-SIM."
* * *

---

## Author Response (AR2)

Dear editor,

The authors would like to thank you for reviewing this manuscript and applaud your attention to detail.

E1) In the spirit of one of the comments by reviewer #1 I would like to see the term "uncertainty quantification" replaced by "uncertainty assessment" wherever it appears.

A1) This change was made throughout the manuscript.

E2) Figure 1 has been changed quite substantially. Now it is not really clear why the upper part of the new version is faded out. Is this the part that will be treated in the follow-on paper? Please clarify (in the figure caption).

A2) The figure caption now reads: "Follow-on work will describe how observation can be post-processed and resampled to reduce the scale gap before model evaluation can be performed."

E3) Figure 2, caption: change b_2-4 into b_1-4 (first line).

A3) Great catch by the editor. The correction was made.

E4) 1st par. of section 3: "radiative scattering transfer" is a bad expression. I think "radiative transfer" suffices, since this expression includes scattering. If you deem the process of scattering should be emphasized here, then change to something like "radiative transfer, in particular scattering". The expression occurs twice and should be changed.

A4) Great suggestion. The expression "radiative scattering transfer" was changed to "radiative transfer" throughout the manuscript.

E5) 1st par of 3.1: "cloud particles backscatter THIS TYPE OF RADIATION the most". I don't know what this type of radiation is. Please reformulate.

A5) The expression was reformulated: "At a lidar wavelength of 532 nm, backscattered power is proportional to total particle cross section per unit volume. Owing to their high number concentrations, despite their small size, cloud particles backscatter radiation of this wavelength the most."

E6) Table 1, green column: I don't understand what "see questionable row" means, in particular since the row labelled "questionable" is empty in the green column.

A6) The expression "see questionable row" was replaced by: "Approximately equal to sum of questionable row: ($\sim 5.2 \quad \pm \quad 0.9$)".

E7) Lines 254-256 is almost exactly repeated in 276-278; the word "is" is missing in 255. Please correct.

A7) Great catch by the editor. This oversight was corrected.

E8) Eq. 29: Check the units. lhs has m/s, but rhs has (m/2)^2.

A8) The editor is correct, for consistency with the units mentioned in the text a square root was added to Eq. 29.

E9) line 880: "going forward" can be deleted.

A9) The expression "going forward" had been deleted.

E10) line 928: What is CFAD (perhaps I missed the definition?).

A10) The authors omitted to include a definition. The text now reads: "reflectivity contoured frequency by altitude diagrams (CFADs)"

[revised manuscript text omitted]

L024
L025
L026